# An interferon-integrated mucosal vaccine provides pan-sarbecovirus protection in small animal models

Chun-Kit Yuen [1,2,6], Wan-Man Wong[1,2,6], Long-Fung Mak[1,2,6], Joy-Yan Lam [1,2], Lok-Yi Cheung[1], Derek Tsz-Yin Cheung[1], Yau-Yee Ng[1], Andrew Chak-Yiu Lee[1,2], Nanshan Zhong[3], Kwok-Yung Yuen [1,2,4,7] & Kin-Hang Kok [1,2,4,5,7] ✉

A pan-sarbecovirus or pan-betacoronavirus vaccine that can prevent current and potential future beta-coronavirus infections is important for fighting possible future pandemics. Here, we report a mucosal vaccine that cross-protects small animal models from sarbecoviruses including SARS-CoV-1, SARS-CoV-2 and its variants. The vaccine comprises a live-but-defective SARS-CoV-2 virus that is envelope deficient and has the orf8 segment replaced by interferon-beta, hence named Interferon Beta Integrated SARS-CoV-2 (IBIS) vaccine. Nasal vaccination with IBIS protected mice from lethal homotypic SARS-CoV-2 infection and hamsters from co-housing-mediated transmission of homotypic virus. Moreover, IBIS provided complete protection against heterotypic sarbecoviruses, including SARS-CoV-2 Delta and Omicron variants, and SARS-CoV-1 in both mice and hamsters. Besides inducing a strong lung CD8 + T cell response, IBIS specifically heightened the activation of mucosal virus-specific CD4 + T cells compared to the interferon-null vaccine. The direct production of interferon by IBIS also suppressed virus co-infection of SARS-CoV-2 in human cells, reducing the risk of genetic recombination when using as live vaccines. Altogether, IBIS is a next-generation pan-sarbecovirus vaccine and warrants further clinical investigations.

Currently, there are ongoing research and clinical trials on various strategies to enhance the effectiveness of COVID-19 vaccines. Scientists are exploring new vaccine platforms such as virus-like particles (VLP), live-attenuated viruses, replicative viral vectors, and multiple receptor binding domain (RBD) protein subunits. The aim is to develop next-generation broadly protective vaccines that will provide optimal protection for the respiratory mucosa by eliciting mucosal immunity[1]. These vaccines are also expected to be effective against different SARS-CoV-2 variants and other coronaviruses. However, the challenge lies in administering these vaccines directly into the respiratory tract, as the current intramuscular injections might not be suitable for this purpose.

SARS-CoV-2, which causes severe respiratory infections in humans, is the third zoonotic coronavirus following the outbreak of SARS-CoV-1 and MERS-CoV. Ideally, a vaccine that can prevent the spread of any future harmful coronaviruses, such as a pan-betacoronavirus or pan-sarbecovirus vaccine, would be preferred. One potential solution to

[1]Department of Microbiology, Li Ka Shing Faculty of Medicine, The University of Hong Kong, Hong Kong SAR, China. [2]Centre for Virology, Vaccinology and Therapeutics, Hong Kong Science and Technology Park, Hong Kong SAR, China. [3]Department of Respiratory Medicine, the First Affiliated Hospital of Guangzhou Medical University, Guangzhou Institute of Respiratory Health, China State Key Laboratory of Respiratory Disease, National Clinical Research Center for Respiratory Disease, Guangzhou 510120, China. [4]State Key Laboratory for Emerging Infectious Diseases, The University of Hong Kong, Hong Kong SAR, China. [5]AIDS Institute, Li Ka Shing Faculty of Medicine, The University of Hong Kong, Hong Kong SAR, China. [6]These authors contributed equally: Chun-Kit Yuen, Wan-Man Wong, Long-Fung Mak. [7]These authors jointly supervised this work: Kwok-Yung Yuen, Kin-Hang Kok. ✉e-mail: khkok@hku.hk

elicit pan-sarbecovirus mucosal immunity is through nasal vaccination with live-attenuated SARS-CoV-2 viruses. Nasal immunization is designed to simulate a natural infection, which is expected to stimulate the protective B and T cell immunity. Additionally, live-attenuated viruses used in this type of immunization express multiple viral proteins, including the spike protein. This approach will likely generate an adaptive immune response targeting antigens shared by different coronaviruses. Recent studies have reported the successful creation of live-attenuated SARS-CoV-2 viruses. For example, Zhang et al. deleted the orf3a-envelope locus of the virus resulting in the ablation of viral replication[2]. Trimpert et al. recoded the viral genome with codons underrepresented in humans to attenuate the virus[3]. Ye et al. introduced a D130A substitution to the viral 2'-O-methyltransferase NSP16 to achieve viral attenuation[4]. All these attenuated viruses showed satisfying animal protection from homologous viral infection and elicited a mucosal immune response. Despite the promising potential use of live-attenuated SARS-CoV-2 viruses as mucosal vaccines, incomplete attenuation and the possibility of generating the replicative revertant virus are major concerns. Therefore, it is crucial to implement measures that can guarantee sufficient attenuation and prevent the generation of the revertant virus before considering the use of live-attenuated SARS-CoV-2 viruses in humans.

The innate antiviral defense against RNA virus infections relies on type-I and III interferons and their signaling[5]. When viral RNA is detected by host pattern recognition receptors like RIG-I-like receptors and Toll-like receptors, it triggers downstream adaptors, kinases, and transcriptional factors, ultimately producing endogenous type-I/III interferons. These interferons function both as autocrine and paracrine agents, limiting viral replication in infected cells and safeguarding neighboring healthy cells from infection. Moreover, the timely production of type-I interferons by infected cells can optimally activate adaptive immune responses, thereby shaping effector and memory T cells[6]. Coronaviruses are known to induce delayed type-I interferon signaling, which is associated with a compromised virus-specific T cell response[7–9]. We and others have shown that several viral proteins of SARS-CoV-2 possess interferon-antagonizing functions[10,11]. A comparative study also suggested that SARS-CoV-2 suppresses type-I and type-III interferon production more robustly than SARS-CoV-1 in an ex-vivo human lung tissue culture model[12]. Collectively, coronaviruses including SARS-CoV-2 evade innate immunity by delaying type-I interferon production, which leads to an impaired adaptive immune response.

In this study, we rationally designed a live-attenuated SARS-CoV-2 virus intended for mucosal immunization. Multiple early-stop codons were introduced into the viral envelope gene, resulting in a strictly single-round viral infection. Notably, an interferon-beta (IFNβ) expression cassette was also inserted into the viral genome. Therefore, the virus would express IFNβ in concert with other viral proteins, mimicking a natural infection with a timely interferon response. One of the benefits of this virus-encoded IFNβ is its ability to act as a secondary safeguard against unexpected co-infection, making it an effective innate antiviral cytokine. The direct production of interferon by IBIS proved effective in suppressing co-infection with the Omicron virus, thereby enhancing the safety of a live virus being used as a nasal vaccine. On the other hand, the timely expression of type-I interferon could also help promote an optimal adaptive immune response. We demonstrated that integrating interferon-beta in IBIS not only induced a strong lung CD8 + T cell response but also specifically heightened the activation of mucosal virus-specific CD4+ T cells compared to the interferon-null vaccine. Nasal vaccination with IBIS resulted in a complete protection against both lethal homotypic SARS-CoV-2 infection and heterotypic sarbecovirus infections. Given these promising results, the current design of IBIS has the potential to be developed into a pan-sarbecovirus vaccine, and further investigation and clinical trials are warranted.

## Results

### Rational design and high-titer generation of an interferon-integrated SARS-CoV-2 vaccine

Live-attenuated SARS-CoV-2 virus is a plausible candidate for mucosal vaccines against SARS-CoV-2 infections, provided that the vaccine's safety is guaranteed. In this study, we rationally designed a mucosal vaccine composed of a highly attenuated SARS-CoV-2 virus, which could infect cells for a single round strictly due to loss of viral envelope protein expression. Three in-frame stop codons were introduced into the envelope gene of the recombinant SARS-CoV-2 genome, resulting in a complete abrogation of the envelope protein expression and hence the abolishment of progeny virus production (Fig. 1a and Supplementary Fig. 1a). To further ensure the attenuation of the virus, the viral genome was integrated with an expression cassette for interferon-beta (IFNβ), which is a well-studied broad-spectrum antiviral cytokine, by replacing orf8 segment in the viral genome. This design is therefore named the Interferon-Beta-Integrated SARS-CoV-2 (IBIS) vaccine. To facilitate vaccine production, we generated an IBIS production VeroE6 cell-line that stably expresses SARS-CoV-2 envelope protein to complement the loss of envelope expression by the IBIS virus. The stable cells harbor an engineered envelope (eE) transgene containing 90 synonymous nucleotide substitutions to minimize homology and prevent the possible recombination between the envelope transgene and the defective viral genome, which could lead to the generation of replicative revertant virus (Supplementary Fig. 1b). Stable single-cell clones were isolated, and the clone Vero A9 was selected for downstream modifications. IBIS expresses the antiviral cytokine IFNβ during replication in the supportive production cells, which would in turn inhibit the viral production. To tackle this, we further abrogated the interferon signaling in Vero A9 cells by knocking out the STAT1 gene using CRISPR-Cas9. Clonal cells were isolated, and the clone Vero A9B21 was selected for vaccine production after confirmation of STAT1 knockout and the abrogation of interferon signaling as indicated by the loss of interferon-stimulated gene expression upon recombinant IFNβ treatment (Supplementary Fig. 1c).

We first generated a version of IBIS with mouse IFNβ for functional characterization in rodent animal models. To confirm the single-round infection of IBIS, parental VeroE6 cells and Vero A9B21 cells were either mock infected, or infected with the IBIS vaccine, a defective SARS-CoV-2 with modified envelope (SARS2-mE), or wildtype SARS-CoV-2 virus at a multiplicity of infection (MOI) 0.1 for 24 h (Fig. 1b). Immunostaining for SARS-CoV-2 nucleoprotein (N) indicated the successful infection and viral antigen expression by both the IBIS and SARS2-mE viruses in both envelope-deficient and envelope-sufficient cells. However, IBIS and SARS2-mE infection was only detected in a few parental VeroE6 cells, contrary to their robust replication in Vero A9B21 as indicated by the N protein staining. Furthermore, parental VeroE6 cells infected with IBIS or SARS2-mE did not produce progeny infectious virus at 48 h post-infection, while both viruses could replicate to the titer of $10^6$ plaque-forming units per milliliter (PFU/mL) in Vero A9B21 cells (Fig. 1c). We further ascertained the single-round infection of IBIS in mouse L929 cells stably expressing human ACE2 (L929-hACE2) (Fig. 1d). Like VeroE6 cells, punctate N protein staining was observed in L929-hACE2 cells infected by IBIS or SARS2-mE at 24 h post-infection, while the wildtype SARS-CoV-2 virus replicated robustly in the cells. IBIS and SARS2-mE expressed viral antigens in the infected cells for at least 24 h post-infection. But these defective viruses were naturally lost later as indicated by the fading of N protein signal at 48–72 h post-infection without causing cytopathic effect (CPE), while the wildtype SARS-CoV-2 caused robust CPE (Supplementary Fig. 2a). Moreover, no infectious virus could be detected after IBIS or SARS2-mE infection of L929-hACE2, as compared to the infection by wildtype virus which replicated to the titer of about $1 \times 10^5$ PFU/mL (Supplementary Fig. 2b). Taken together, IBIS is a highly attenuated virus that could only infect for a single round and transiently express

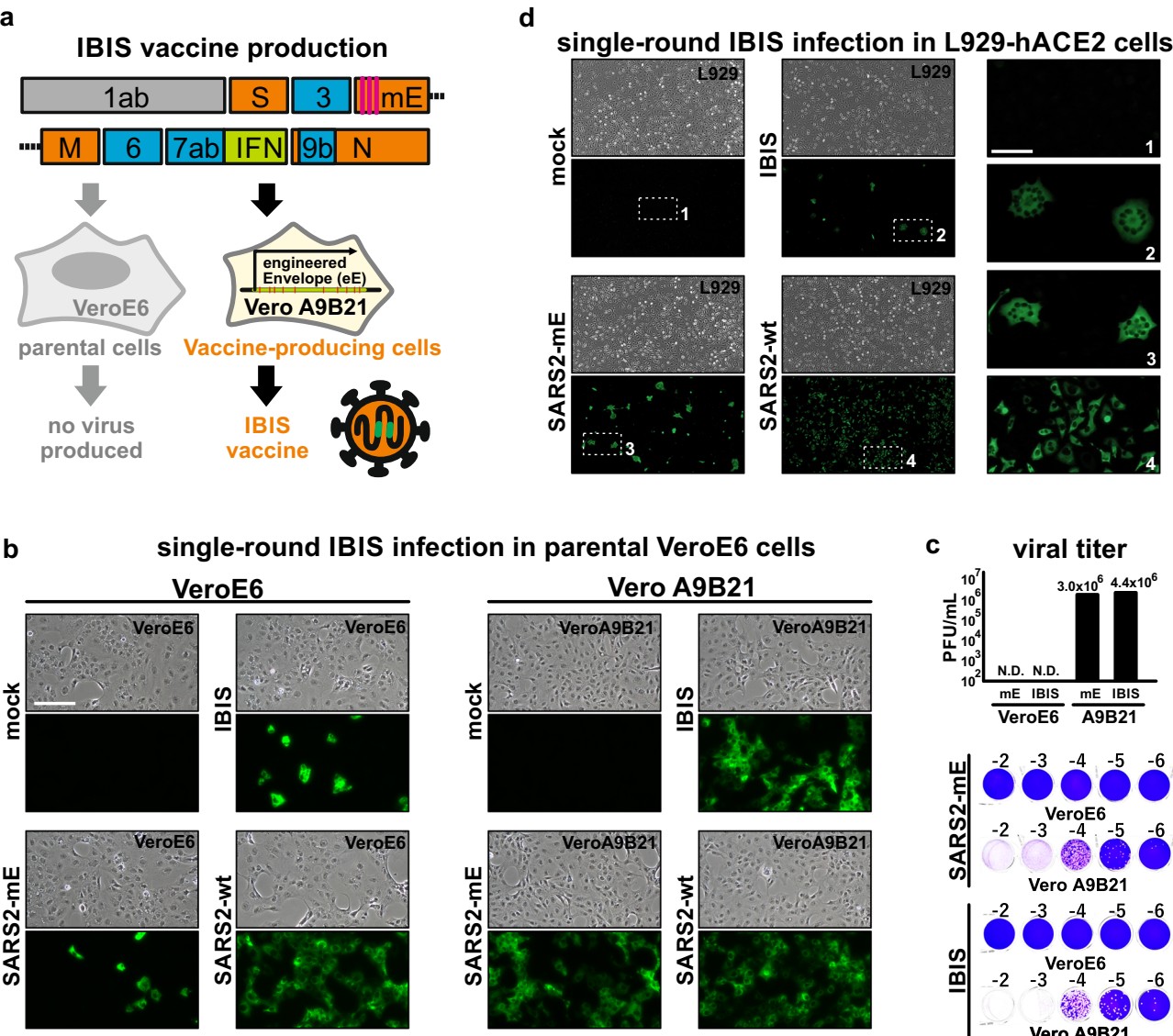

**Fig. 1 | Rational design and high-titer generation of an interferon-integrated SARS-CoV-2 vaccine. a** Schematic diagram of IBIS design. IBIS is a live-but-defective SARS-CoV-2 virus. It contains a modified envelope (mE) gene that has three early stop codons (pink lines) introduced to abrogate envelope protein expression. An open-reading frame coding for interferon-beta (IFNβ) was also inserted in place of orf8 (green box). A version of IBIS having mouse IFNβ (mIFNβ) was produced for mouse and hamster experiments. The virus was rescued and propagated in the production cell Vero A9B21, which stably expresses an engineered Envelope transgene and has the STAT1 gene knocked out.
**b** Immunofluorescence staining showing IBIS multiple-round infection in Vero A9B21 but not in parental VeroE6 cells. Parental VeroE6 (left panel) or Vero A9B21 cells (right panel) were either mock-infected or infected with IBIS, SARS-CoV-2 virus harboring mE but without mIFNβ transgene (SARS2-mE), or wildtype SARS-CoV-2 virus (SARS2-wt; ancestral) at MOI 0.1. Twenty-four hours post-infection, cells were PFA-fixed and immunostained for SARS-CoV-2 N protein. **c** Plaque assay of SARS2-mE virus and IBIS in parental VeroE6 and Vero A9B21 cells. SARS2-mE and IBIS viruses were propagated in Vero A9B21, and titered by plaque assay using both parental VeroE6 cells and Vero A9B21 cells. Upper panel shows the titer in plaque-forming unit per milliliter (PFU/mL). Lower panel shows the representative plaque images. **d** Immunofluorescence staining showing IBIS single-round infection in mIFNβ-responsive cells. Murine fibroblasts L929 stably expressing a human ACE2 (hACE2) transgene were either mock-infected or infected with IBIS, SARS2-mE, or wildtype SARS-CoV-2 virus at MOI 0.1 (left and middle panels). Cells were PFA-fixed at 24 h post-infection and stained for SARS-CoV-2 N protein. Enlarged images of the selected regions are shown in right panels. Scale bar = 50 μm. The experiments were repeated for 3 times with similar results obtained. Source data are provided as a Source Data file.

viral antigens. The vaccine can only be produced in the envelope-expressing STAT1-knockout Vero A9B21 cells.

**In-vitro and in-vivo expression of interferon-beta by IBIS**
An expression cassette of the antiviral cytokine IFNβ was inserted into the IBIS viral genome to enhance the safety of the vaccine further. To confirm the expression of functional IFNβ by IBIS and the genome stability, IBIS was passaged ten times in Vero A9B21. The amount of IFNβ protein in the viral supernatant of passage 1, 6 and 10 was quantitated by ELISA (Fig. 2a). Supernatant from all three passages showed a similar concentration of IFNβ protein (>1500 pg/mL), indicating that IBIS could successfully express IFNβ; and the IFNβ-expression cassette could be stably maintained in the viral genome. The antiviral function of the IFNβ produced was further testified by VSV-GFP inhibition bioassay (Fig. 2b). L929 cells were treated with 1:1000 diluted passage 1, 6 and 10 viral supernatant, followed by infection with a GFP-expressing vesicular stomatitis virus (VSV-GFP). The supernatant of all three passages could quell VSV-GFP infection,

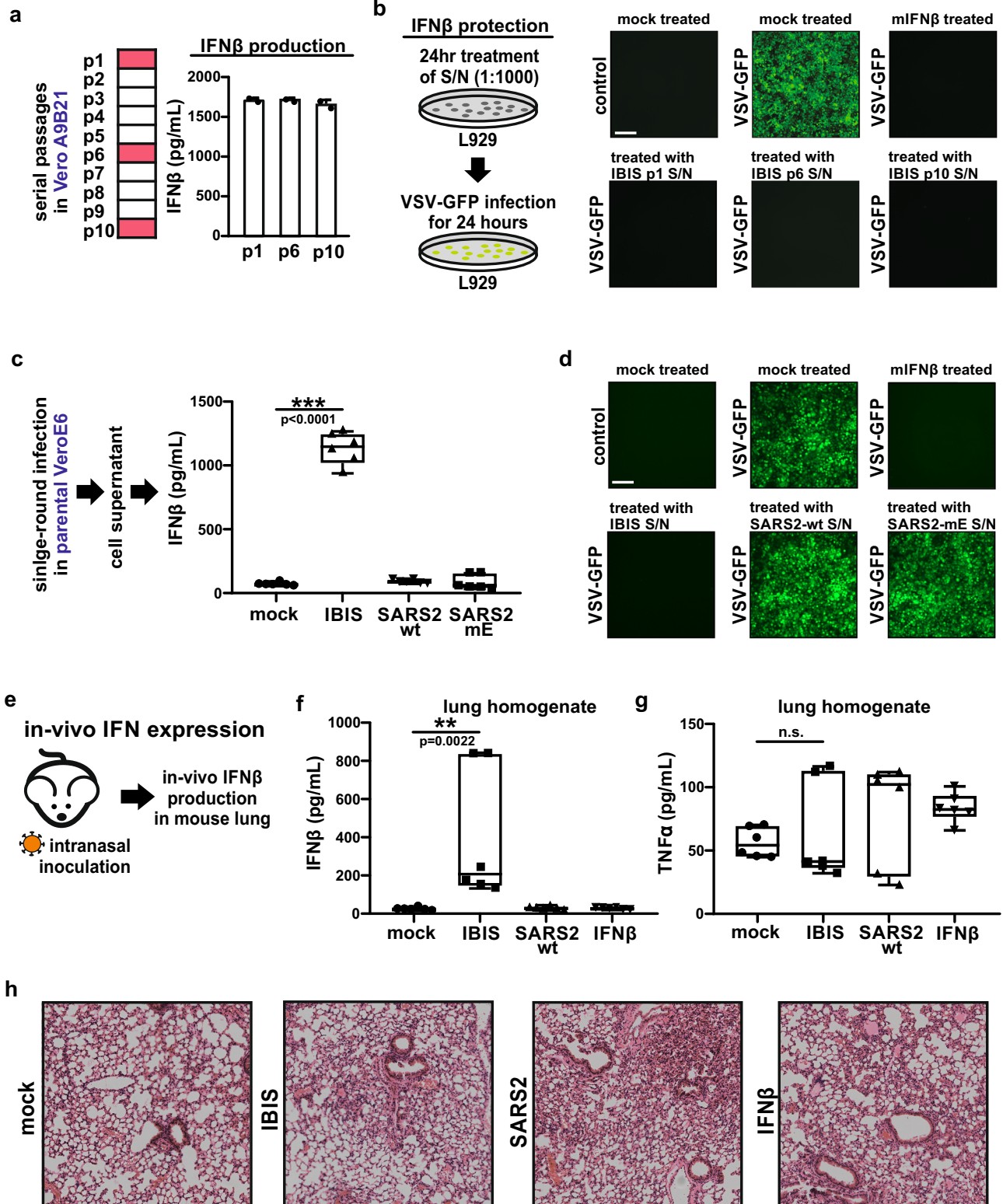

suggesting that the IFNβ encoded by IBIS possesses antiviral functions. To further test if sufficient IFNβ is produced upon single-round IBIS infection, parental VeroE6 cells were infected by IBIS, SARS2-mE or wildtype SARS-CoV-2 virus at MOI 0.1. Twenty-four hours post-infection, the supernatant was harvested for IFNβ ELISA and VSV-GFP inhibition assay. Although IBIS could not replicate for multiple rounds in parental VeroE6 cells, a significant amount of IFNβ was secreted after a single-round IBIS infection (Fig. 2c). Moreover, only the

supernatant of VeroE6 cells infected with IBIS, but not those with SARS2-mE nor wildtype SARS2, could inhibit VSV-GFP infection, suggesting that the protection was mediated by the IBIS-produced IFNβ (Fig. 2d).

We further confirmed the production of IFNβ in vivo by intranasally inoculating $1\times10^6$ PFU of IBIS into K18-hACE2 transgenic mice (Fig. 2e). A significant amount of mouse IFNβ protein could be detected in the lung homogenates of IBIS-infected mice at 24 h post-

**Fig. 2 | In-vitro and in-vivo expression of interferon-beta by IBIS. a** Serial passaging of IBIS and quantitation of IFNβ by ELISA. IBIS was serially passaged ten times in Vero A9B21 cells. Supernatant was harvested when cytopathic effect (CPE) was >80%. IFNβ in the viral supernatant of passage 1, 6 and 10 was quantitated by ELISA. n = 2. Error bar=S.D. **b** Interferon bioassay showing functional IFNβ produced by IBIS. L929 cells were either mock-treated, treated with 1000 IU/mL recombinant mouse IFNβ, or with 1:1000 diluted IBIS viral supernatant (S/N) of passage 1, 6 and 10. At 24 h post-treatment, cells were subjected to GFP-reporter vesicular stomatitis virus (VSV-GFP) infection. GFP signal was observed at 24 h post-infection under a fluorescence microscope. Scale bar=300 μm. The experiments were repeated for 3 times with similar results obtained. **c, d** Quantitation of IFNβ and determination of IFNβ activity after single-round IBIS infection. Parental VeroE6 cells were either mock-infected, or infected with IBIS, wildtype SARS-CoV-2 (SARS2-wt; ancestral) or SARS2-mE at MOI 0.1. Culture supernatant was collected at 24 h post-infection. The amount of IFNβ was quantitated by ELISA (**c**). n = 6. Center line, median; box limits, upper and lower quartiles; whiskers, minima and maxima. The presence of functional interferon in the undiluted supernatant was further testified by VSV-GFP interferon bioassay in L929 cells (**d**). **e** Expression of IFNβ by IBIS in vivo. Six to ten week-old male K18-hACE2 transgenic mice were either intranasally inoculated with PBS, IBIS ($1\times10^6$ PFU), SARS2-wt ($1\times10^3$ PFU) or recombinant mouse IFNβ ($1\times10^5$ IU). At 24 h post-infection/treatment, mouse lungs were harvested and homogenized in PBS. The amount of mouse IFNβ (**f**) and TNFα (**g**) in the lung homogenate was quantitated by ELISA. n = 6. Center line, median; box limits, upper and lower quartiles; whiskers, minima to maxima. Mouse lungs were also PFA-fixed for sectioning and hematoxylin and eosin (H&E) staining (**h**). Two-sided unpaired student t test was used in statistical analysis of cell culture experiment. Two-sided non-parametric Mann-Whitney test was used in statistical analysis of animal experiments. \*\*, p < 0.01; \*\*\*, p < 0.001; n.s., not significant. Source data are provided as a Source Data file.

inoculation (Fig. 2f). In line with our in vitro data (Fig. 2c, d) and previous reports demonstrating that coronaviruses induce delayed type I interferon response[7–9], wildtype SARS-CoV-2 infection did not robustly induce IFNβ in the infected mouse lungs at 24 h post-infection. As expected, no mouse IFNβ protein could be detected in the lungs of mice at 24 h after intranasal instillation of $1\times10^5$ IU recombinant mouse IFNβ, which might indicate the rapid turnover of IFNβ in the lungs. Importantly, IP-10, which is an interferon-inducible chemokine, was specifically induced by IBIS but not SARS2-mE in the bronchoalveolar lavage (BAL) of mice at 18 h post-inoculation (Supplementary Fig. 3a–c). Single cell analysis further showed that IBIS specifically induced subsets of immune cells displaying interferon-inducible gene (ISG) signature (Supplementary Fig. 3d–f), suggesting that the IFNβ expressed by IBIS is functional in vivo. Upon vaccination with IBIS, no induction of inflammatory cytokine TNFα could be detected (Fig. 2g) in the lungs of immunized mice, of which the alveolar structure remained intact (Fig. 2h and Supplementary Fig. 4a). The IBIS-vaccinated mice also did not show any body weight loss nor disease symptoms over 14 days of monitoring (Supplementary Fig. 4b–d).

### Intranasal IBIS vaccination protects mice against lethal SARS-CoV-2 infection

We next examined the antibody response and protective efficacy of IBIS vaccination on K18-hACE2 transgenic mice. Mice were vaccinated by intranasal inoculation of two doses ($1\times10^6$ PFU per dose) of IBIS at an interval of 14 days. The vaccinated mice were bled at time points for serum collection, followed by a lethal SARS-CoV-2 (ancestral strain) challenge at 28 days post-vaccination (Fig. 3a). A single dose of IBIS was able to elicit neutralizing antibodies in all vaccinated mice at 14 days post-vaccination. Impressively, a booster given at day 14 could further raise the serum-neutralizing antibody titer to about 2 logs higher, with average $FRNT_{50}$ and $FRNT_{75}$ titers approaching $10^4$ (Fig. 3b). A high level of anti-RBD IgG antibody was elicited to a comparable level of two-dose mRNA BioNTech (BNT) vaccination (Fig. 3c). In line with our in vitro findings showing that IBIS encodes a detectable level of SARS-CoV-2 nucleoprotein (Fig. 1b, d), IBIS induced nucleoprotein-specific IgG antibodies, which is lacking in immunization with mRNA vaccine that encodes SARS-CoV-2 Spike protein solely (Fig. 3d).

At 28 days post-vaccination, the mice were lethally challenged by intranasal inoculation of the SARS-CoV-2 virus. All IBIS-vaccinated mice survived from the lethal infection and did not show body weight loss (Fig. 3e, f). No infectious virions could be detected in the lungs of vaccinated mice at day 2 post-infection (Fig. 3g, middle column). Besides, no virus could be detected at 2 days after inoculating $1\times10^6$ PFU of IBIS, indicating the inability of IBIS to replicate for multiple rounds in vivo (Fig. 3g, right column). Immunofluorescence staining against SARS-CoV-2 nucleoprotein showed that no viral antigens could be detected in the lungs of IBIS-vaccinated mice at 2 days post-infection, contrary to the robust viral replication in the entire lungs of mock-vaccinated mice (Fig. 3h, i and Supplementary Fig. 5). Taken together, IBIS vaccination induced high neutralizing antibody titer and antibodies against both spike and nucleoprotein, protecting vaccinated mice from lethal SARS-CoV-2 infection and morbidity.

### IBIS vaccination impedes SARS-CoV-2 transmission in a hamster co-housing model

Golden Syrian hamsters have been shown susceptible to SARS-CoV-2 infections[13]. Despite being a non-lethal model, pathophysiology in hamsters resembles more closely to human infections when compared to other small animal models[14]. We therefore further characterized our IBIS vaccine in preventing SARS-CoV-2 transmission using a hamster co-housing model (Fig. 4a). Hamsters were vaccinated with two doses ($3\times10^6$ PFU per dose) of IBIS at a 14-day interval, followed by co-housing with SARS-CoV-2-infected hamsters. In line with mouse immunization, IBIS induced a robust antibody response against both SARS-CoV-2 RBD and nucleoprotein in vaccinated hamsters (Fig. 4b, c). A similar boosting effect was also observed in hamsters, as demonstrated by the fact that a booster given at day 14 could enhance neutralizing antibody titer to about 1.5 logs higher, with high $FRNT_{50}$ and $FRNT_{75}$ titers ranging from $10^3$ to $10^4$ (Fig. 4d). To evaluate the efficacy of IBIS vaccination in preventing SARS-CoV-2 transmission, naïve and IBIS-vaccinated hamsters were co-housed with pre-infected (ancestral strain) index hamsters; and then separated at one day after co-housing. All IBIS-vaccinated hamsters did not lose body weight after co-housing, while the naïve hamsters lost more than 10% of body weight (Fig. 4e). At day 5 post-co-housing, lung sections of naïve hamsters showed patchy damaged areas with the loss of normal alveolar structure and nucleoprotein immunofluorescence staining (Fig. 4h and Supplementary Fig. 6). On the contrary, IBIS vaccination protected hamsters from lung damage post-co-housing. No nucleoprotein could be detected in the hamster lungs (Fig. 4i and Supplementary Fig. 6). Moreover, no infectious virus could be recovered from the lungs and nasal turbinate of IBIS-vaccinated hamsters, while mock-vaccinated hamsters showed a high titer of infectious virus in the nasal turbinate at day 2 and lungs at both day 2 and day 5 post-co-housing (Fig. 4f, g and Supplementary Fig. 7a, b). Viral RNA transcript level was also 3-5 logs lower in the respiratory tract tissues of vaccinated hamsters (Supplementary Fig. 7c–h). We further tested the minimal dose of IBIS that could protect hamsters from SARS-CoV-2 infection-mediated body weight loss. To avoid masking of subtle differences by the booster dose, hamsters were immunized with a single dose of serially tenfold diluted IBIS ranging from $3\times10^2$ to $3\times10^6$ PFU, followed by an intranasal SARS-CoV-2 infection at 14 days post-vaccination. Surprisingly, a single dose of $3\times10^2$ PFU IBIS was sufficient to protect the vaccinated hamsters from weight loss, while hamsters vaccinated with $3\times10^5$ PFU IBIS showed the highest degree of weight gain at day 14 post-infection (Fig. 4j).

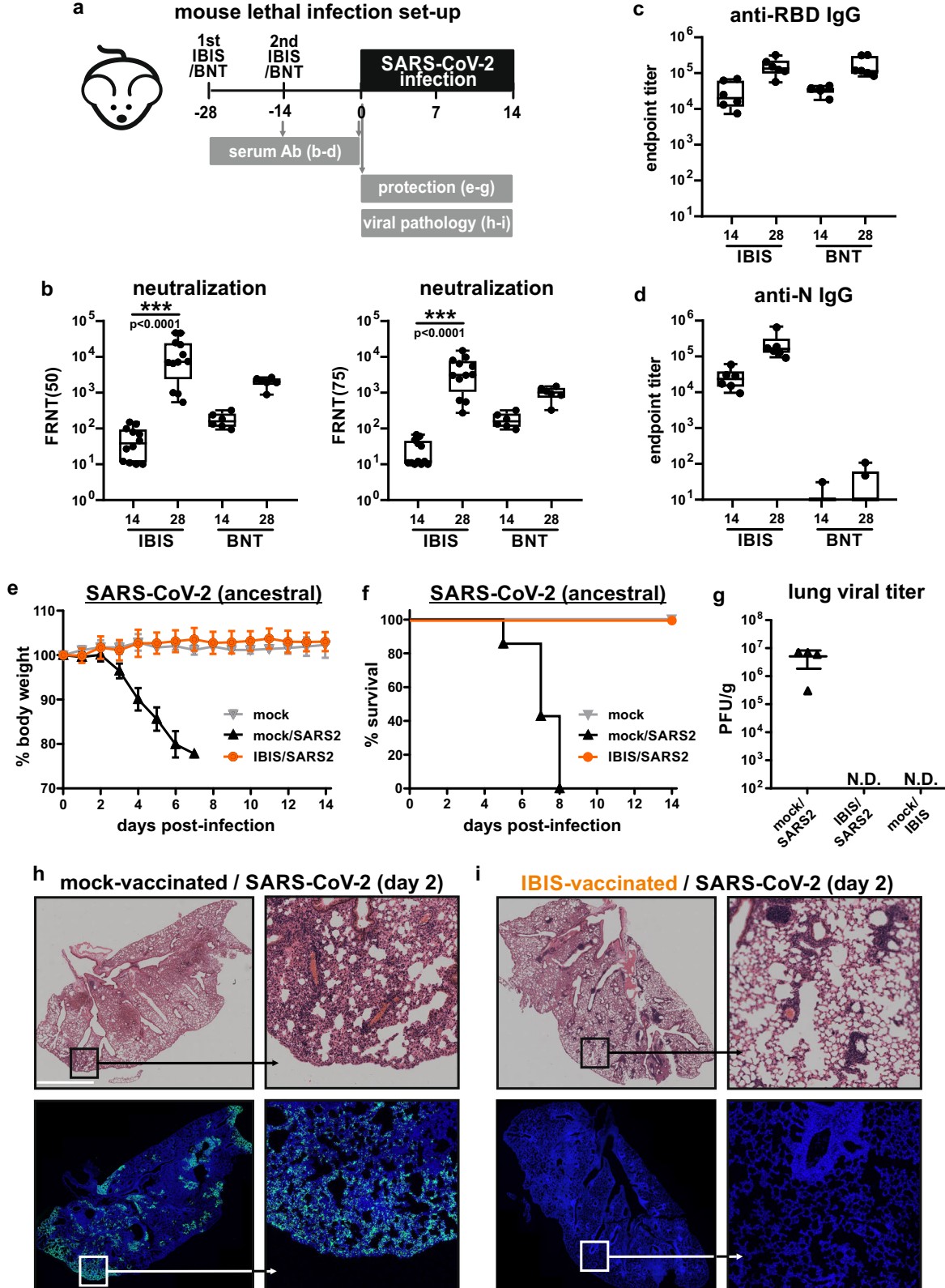

## IBIS vaccination prevents Delta and Omicron variant infection and transmission

The SARS-CoV-2 virus has been evolving since its first emergence in late 2019. Variants of concern including Alpha, Beta, Gamma, Delta and Omicron have emerged along the pandemic and raised concerns about their escape from immunity elicited by the currently used vaccines. To evaluate the protective efficacy of IBIS against these variants, we infected

IBIS-vaccinated K18-hACE2 transgenic mice with either Delta or Omicron (sub-lineage BA.1) virus and quantitated the lung viral titer. As expected, no infectious Delta or Omicron virus could be detected in the vaccinated mouse lungs at both day 2 and day 5 post-infection (Fig. 5a, b). The viral RNA transcript level was also lower in IBIS-vaccinated mouse lungs and nasal turbinate (Supplementary Fig. 8–9). In addition to BA.1, we further confirmed that IBIS is also effective against the more recent Omicron

**Fig. 3 | IBIS vaccination induces a potent antibody response and protects mice against lethal SARS-CoV-2 infection. a** Schematic timeline of vaccination and viral challenge. Six to ten week-old male K18-hACE2 transgenic mice were vaccinated with 2 doses of either intranasal IBIS vaccine ($1\times10^6$ PFU/dose; n = 12) or with intramuscular BioNTech (BNT) BNT162b2 mRNA vaccine (1 μg/dose; n = 6) on day −28 and day −14. Mouse sera were collected at 14 days after each dose of vaccination. Neutralizing antibody titer of the sera was determined by focus-reduction neutralization test with 50% reduction cut-off ($FRNT_{50}$) or 75% cut-off ($FRNT_{75}$) against authentic SARS-CoV-2 (ancestral) virus (**b**). The titer of IgG antibodies specific to SARS-CoV-2 spike-receptor binding domain (RBD) and nucleoprotein (N) was determined by ELISA (**c, d**); n = 6 per group. Center line, median; box limits, upper and lower quartiles; whiskers, minima and maxima. On day 0, mock- and IBIS-vaccinated mice were lethally challenged by intranasal inoculation of SARS-CoV-2 virus (ancestral) ($1\times10^3$ PFU). Body weight change (**e**) and survival (**f**) of the infected mice were monitored for 14 days. Mock-infected, n = 3; Mock-vaccinated/SARS-CoV-2 infected, n = 7; IBIS-vaccinated/SARS-CoV-2 infected, n = 6. Lungs of mock-vaccinated mice (n = 4) and IBIS-vaccinated mice (n = 5) were harvested at day 2 post-infection to determine the lung viral load by plaque assay (**g**). A group of mice were also vaccinated with $1\times10^6$ PFU of IBIS and then sacrificed at day 2 post-vaccination to confirm the absence of IBIS productive infection in vivo (n = 3). Data are presented as mean ± standard deviation (SD). PFU/g, plaque-forming unit per gram of tissue; N.D., not detected. **h–i** Histology of infected mouse lungs. Mock- or IBIS-vaccinated mice challenged by SARS-CoV-2 (ancestral) ($1\times10^3$ PFU) were sacrificed at 2 days post-infection. Lungs of the infected mice were harvested and PFA-fixed for histological examination by H&E staining (upper panels) and immunofluorescence (IF) staining for SARS-CoV-2 N (green). Nuclei were counterstained by Hoechst 33258 (blue). PBS-vaccinated, n = 4; IBIS-vaccinated, n = 5. White scale bar=0.2 cm. Two-sided non-parametric Mann-Whitney test was used to calculate statistical significance. ***, p < 0.001. Source data are provided as a Source Data file.

sub-lineages BA.2, BA.4.1 and BA.5.2, reducing the lung infectious viral titer to an undetectable level (Fig. 5c). Furthermore, IBIS vaccination was able to thwart Delta and Omicron transmission in the hamster co-housing model. Vaccinated hamsters were protected from weight loss caused by Delta transmission (Fig. 5d). No infectious virus could be detected along the entire respiratory tract from the nasal turbinate, and trachea, to the lungs of all IBIS-vaccinated hamsters (Fig. 5e). The viral RNA transcript level was also reduced by about 3 logs or more in these respiratory tissues (Supplementary Fig. 10). In line with previous studies reporting the lower pathogenicity of the Omicron variant in hamsters and mice[15–18], minimal body weight decrease and quick viral clearance were observed in the index hamsters infected with the Omicron-BA.1 virus (Fig. 5f, g, purple). Nonetheless, infectious virus was absent in the respiratory tracts of all IBIS-vaccinated hamsters (Fig. 5g, orange), in which the viral RNA transcript level was also reduced compared with the mock vaccinated co-housed hamsters (Supplementary Fig. 11).

### IBIS vaccination elicits heterotypic protection against SARS-CoV-1 infection

Mucosal immunization and pan-coronavirus cross-protection are the two directions toward next-generation SARS-CoV-2 vaccines. IBIS is a mucosal vaccine designed for intranasal administration. Given the complete protection against ancestral, Delta and Omicron variants, we further questioned whether IBIS could be efficacious to other closely related coronaviruses. Surprisingly, IBIS vaccination completely protected K18-hACE2 transgenic mice from lethal SARS-CoV-1 infection. No body weight loss was observed in all IBIS-vaccinated mice, resulting in 100% survival; while mock-vaccinated mice showed a > 10% decrease in body weight and all succumbed to the infection (Fig. 6a, b). Moreover, IBIS vaccination reduced lung viral titer by about 2.5 logs at day 2, and to an undetectable level at day 5 post-infection (Fig. 6c). No infectious virus could be recovered from the nasal turbinate of the vaccinated mice at both day 2 and day 5 post-infection (Fig. 6d). A similar observation could be made in the hamster infection model, in which vaccinated hamsters were intranasally inoculated with SARS-CoV-1 virus. While mock-vaccinated hamsters had high and sustained lung viral titer, no infectious virus was detected in the lungs of IBIS-vaccinated hamsters at both day 2 and day 5 post-infection (Fig. 6e). Although infectious virus at a lower but detectable level was present in the trachea and nasal turbinate of the vaccinated hamsters at day 2, it dropped to undetectable/marginally detectable levels at day 5 post-infection (Fig. 6f, g). Taken together, both the mouse and hamster infection models suggested that the IBIS vaccine can cross-protect against the heterotypic SARS-CoV-1 (Fig. 6h).

### IBIS vaccination specifically enhances mucosal polyfunctional CD4 + T cell activation

Type-I interferons have been shown to facilitate CD4 + T cell expansion and survival[19–21], and inhibit regulatory T cell-mediated suppression of antigen-specific CD4 + T cells[22,23]. It has also been suggested that the delayed type-I interferon signaling during coronavirus infections dysregulates the virus-specific T cell response[7]. Therefore, the early expression of IFNβ by IBIS might be beneficial for inducing an optimal SARS-CoV-2-specific T cell response. To address this issue, the IBIS vaccine was compared with its counterpart, which does not contain the IFNβ expression cassette replacing orf8 (SARS2-mE) (Fig. 7a). K18-hACE2 transgenic mice were vaccinated with two doses of either IBIS or SARS2-mE at a 14-day interval. At day 7 post-second-dose of vaccination, immune cells from dissociated lungs and bronchoalveolar lavage (BAL) were isolated for SARS-CoV-2 spike-specific T cell assays. Vaccination with either IBIS or SARS2-mE strongly activated antigen-specific CD8 + T cells in lungs and BAL (upper panels of Fig. 7d, e; Supplementary Fig. 12). Approximately 75% of which were bifunctional or multifunctional T cells (lower panels of Fig. 7d, e). Specifically, the antigen-specific activation of CD4 + T cells in both lungs and BAL was induced more potently in the IBIS-vaccinated group than in the SARS2-mE group (upper panels of Fig. 7b, c). In particular, the percentage of IFNγ + CD4 + T cells and TNFα + CD4 + T cells in BAL reached 11%-37% in the IBIS-vaccinated group, which was significantly higher than that in the SARS2-mE-vaccinated group (ranging from 1%-14%) (upper panel of Fig. 7b). An enhancement of CD4 + T cell activation could also be observed in the spleen (Supplementary Fig. 13). The presence of interferon in the vaccine also promoted the CD4 + T cells from monofunctional to bifunctional/multifunctional (lower panels of Fig. 7b, c). In addition to the potent CD8 + T cell response, our data suggest that integration of IFNβ in the IBIS vaccine could further help induce a high level of antigen-specific CD4 + T cell response.

### IBIS vaccination slightly enhances authentic virus neutralization

In addition to the T cell response, subclasses of spike-specific antibodies generated in mice vaccinated with IBIS or SARS2-mE were determined and compared. Groups of mice were vaccinated for two doses and the serum antibodies were qualitatively and quantitatively examined (Fig. 8a). Both IBIS and SARS2-mE elicited a high level of anti-spike RBD IgG1, IgG2a and IgG2b (Fig. 8b–d). Although no significant difference in serum anti-spike RBD IgG levels was observed between the IBIS group and the SARS2-mE group, IBIS induced higher neutralizing power in an authentic virus neutralization assay (Fig. 8e).

### Single-round infection of human IBIS suppresses co-infection of Omicron-BA.2

In this study, we generated the IBIS vaccine that expresses mouse IFNβ for in vivo characterization in mouse and hamster models. However, IFNβ is highly species-specific in a way that mouse IFNβ cannot activate interferon signaling in humans. To help translate our IBIS vaccine to human application, we created a human version of IBIS (hIBIS) that encodes human IFNβ. The hIBIS was successfully generated using the STAT1 knockout Vero A9B21 cells and could

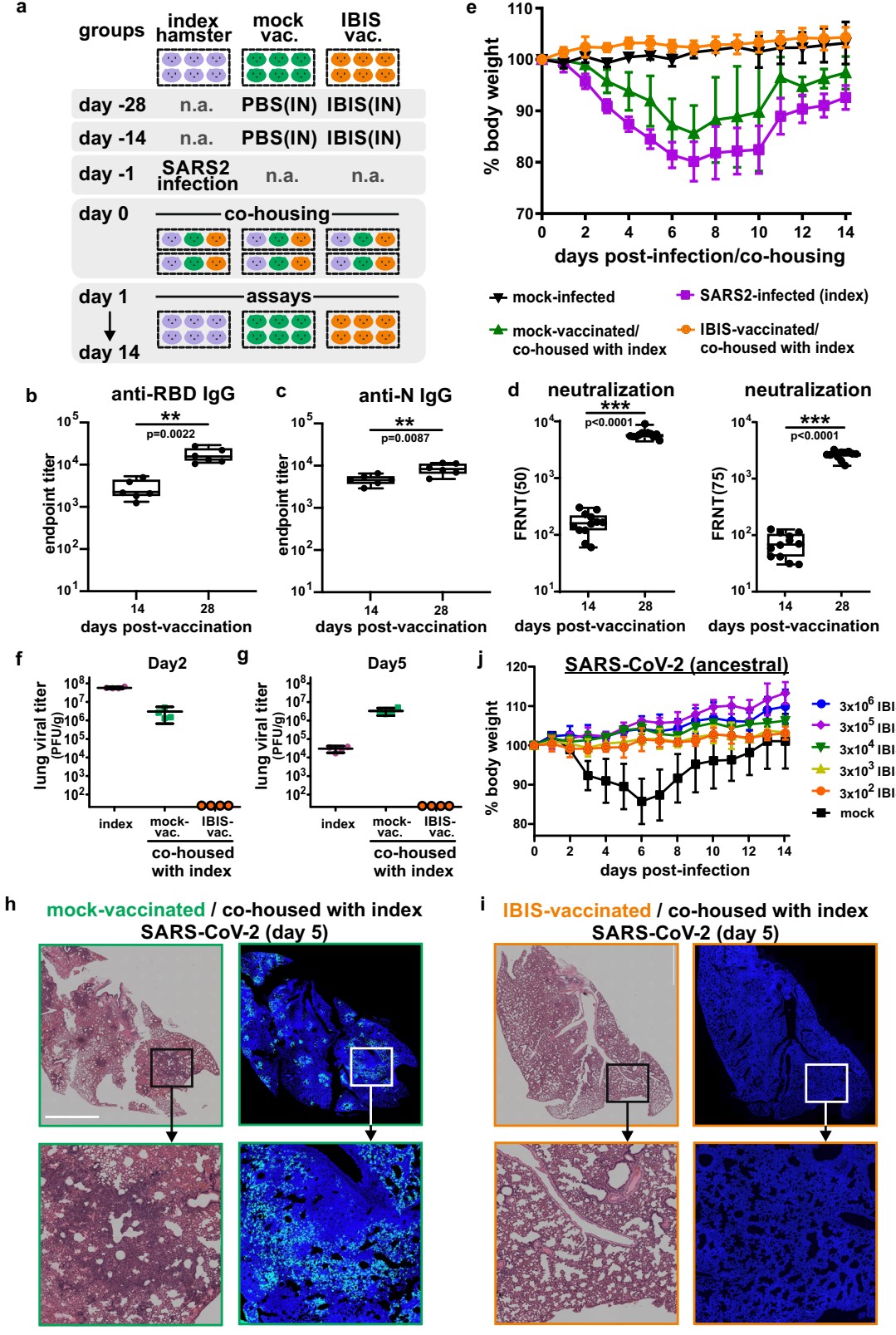

grow to a titer comparable to mouse IBIS. IFNβ is a broad-spectrum antiviral cytokine that can effectively combat an array of viruses. Therefore, the IFNβ expressed by IBIS should inhibit any co-infecting viruses including SARS-CoV-2. We mimicked the co-infection of IBIS with replicative SARS-CoV-2 using an in vitro model, in which A549-hACE2-hTMPRSS2 cells were co-infected with both hIBIS and Omicron-BA.2 (Fig. 9a). Pre-immunized cells with

hIBIS before Omicron-BA.2 infection resulted in more than 3-log reduction of infectious viral titer from 3×10⁵ PFU/mL to 2×10² PFU/mL; while post-treating cells that were already Omicron-BA.2 infected could also reduce the viral titer by more than 1.5 logs (Fig. 9b). In case a vaccinee acquires natural SARS-CoV-2 infection near the time of vaccination, the IBIS-expressed IFNβ could help suppress the growth of the co-infecting virus, minimizing the chance

**Fig. 4 | IBIS vaccination impedes SARS-CoV-2 transmission in hamsters.**
**a** Schematic diagram of hamster vaccination and viral challenge schedule. Six to ten week-old female Golden Syrian hamsters were intranasally vaccinated with 2 doses of IBIS vaccine ($3×10^6$ PFU) on day -28 and day −14, respectively (similar to Fig. 3a). On day −1, index hamsters (purple) were infected by intranasal inoculation of SARS-CoV-2 (ancestral) ($1×10^3$ PFU). One day after infection (day 0), the index hamsters were co-housed with PBS-vaccinated (green) and IBIS-vaccinated (orange) hamsters for 24 h, and then separated. Sera of vaccinated hamsters were harvested at 14 days after each dose of vaccination for the determination of RBD- or N-specific IgG antibody titer (**b**, **c**); n = 6 per group. Neutralizing antibody titer of the sera was determined by $FRNT_{50}$ and $FRNT_{75}$ against authentic SARS-CoV-2 (ancestral) virus (**d**); n = 12 per group. Center line, median; box limits, upper and lower quartiles; whiskers, minima and maxima. Body weight of hamsters was monitored for 14 consecutive days post-infection/co-housing and data are presented as mean ± SD (**e**); n = 6 per group. Lungs of index and co-housed hamsters were harvested at day 2 and day 5 post-infection/co-housing for the determination of lung viral titer by plaque assay (**f**, **g**); n = 4 per group. Data are presented as mean ± SD. The lungs were also PFA-fixed for histological examination by H&E staining (left panels) and IF staining for SARS-CoV-2 N (green) (right panels) at day 5 post-infection/co-housing. Nuclei were counterstained by Hoechst 33258 (blue) (**h**, **i**); n = 4 per group. White scale bar=0.5 cm. To determine the lower effective dose of IBIS, six to ten week-old female hamsters were intranasally vaccinated with a single dose of serially tenfold diluted IBIS ranging from $3×10^2$ PFU to $3×10^6$ PFU per hamster. At 14 days post-vaccination, the hamsters were intranasally challenged with $1×10^3$ PFU of SARS-CoV-2 (ancestral) virus. Body weight change was monitored for 14 days and data are presented as mean ± SD; n = 3 per group (**j**). Two-sided non-parametric Mann-Whitney test was used to calculate statistical significance. ***, p < 0.001. Source data are provided as a Source Data file.

of viral recombination and generation of revertant capable of productive infection.

## Discussion

The COVID-19 pandemic lasted for three years since early 2020. Although SARS-CoV-2 no longer constitutes a public health emergency of international concern as defined by WHO, it continues to circulate globally as the less deadly Omicron variants and causes morbidity. In the past three years, country-wise massive vaccination has played a crucial role in setting back the viral spreading, yet it is also now clear that next-generation vaccines are needed to optimally protect the respiratory mucosa from the viral infection. Ideally, the new vaccine should also provide broad-spectrum protection against coronaviruses beyond SARS-CoV-2. In this study, we report the generation of a COVID-19 mucosal vaccine named IBIS. Besides the protection against SARS-CoV-2 ancestral strain and the Delta and Omicron variants of concern, the merit of IBIS is further highlighted by its promising cross-protection against SARS-CoV-1 virus in two rodent models, which may suggest that it could potentially be a pan-sarbecovirus mucosal vaccine.

Although the envelope protein may not be essential for SARS-CoV-1 and mouse hepatitis virus (MHV)[24,25], a recent study suggested that it is required for SARS-CoV-2[2]. Deletion of envelope causes assembly defect of SARS-CoV-2 and results in a complete abrogation of multiple-round viral infection. In line with this study, IBIS or SARS-mE virus never showed plaque formation in parental VeroE6 cells nor caused cytopathic effect in susceptible cells (Fig. 1b–d and Supplementary Fig. 2). Also, no infectious virus could be recovered from the lungs of mice at two days after intranasal inoculation with $1×10^6$ PFU of IBIS (Fig. 3g, right column). Moreover, the trans-complementation of VeroE6 cells with an envelope transgene rescued IBIS replication, indicating the attenuation is due to the loss of envelope expression. To abrogate envelope expression of IBIS, we introduced three early nonsense mutations instead of deleting the entire envelope gene to minimize the disruption to the IBIS genome, as the latter could potentially lead to genome instability. The multiple stop codons in the rescued IBIS virus could be maintained along serial passaging in the supportive Vero A9B21 cells, resulting in single-round infection and the absence of progeny infectious virions produced both in vitro and in vivo.

To insert an IFNβ expression cassette in the IBIS genome, we aimed to replace one of the ORFs in the SARS-CoV-2 genome with the cassette so that the IFNβ expression is driven by an authentic transcriptional regulatory sequence (TRS) of the virus. Although the SARS-CoV-2 virus was predicted to encode for more than 25 viral proteins, not all of them have been experimentally proven to be expressed at a physiologically relevant level during infection. We previously reported that orf8 is a secretory protein that could be detected in COVID-19 patient sera by LC-MS at 90% protein coverage[26]. We and others also showed that orf8 is an immunogenic protein that elicits an orf8-specific antibody response in human infections[26,27]. These observations point to the fact that TRS of orf8 is functional in driving the expression of a secretory protein at a reasonable level. Moreover, a study during the early stage of pandemic reported a cluster of authentic orf8-deleted viruses circulated in Southeast Asia, indicating that orf8 is dispensable for SARS-CoV-2 replication[28]. Also, orf8 has been shown to impair antigen presentation of infected cells by down-regulating cell surface expression of major histocompatibility complex class I (MHC-I)[29]. Thereupon, we replaced orf8 with IFNβ; and the recombinant IBIS virus that expresses detectable and functional IFNβ protein could be successfully rescued (Fig. 1–2 and Supplementary Fig. 3).

In this study, we showed that IBIS vaccination protected K18-hACE2 transgenic mice from lethal SARS-CoV-2 infection (Fig. 3e–i). As revealed by the histology of the infected mouse lungs, characteristic dark spots were present in all lung samples exclusively in the IBIS-vaccinated group (Fig. 3i and Supplementary Fig. 14a). Immunostaining showed that these spots were cell aggregates that contained B cells and T cells (Supplementary Fig. 14b–d). These densely packed lymphoid aggregates located along large airways and blood vessels highly resembled the characteristic features of inducible Bronchus-Associated Lymphoid Tissues (iBALT), which have been shown beneficial for fighting acute respiratory viral infections including influenza and SARS coronavirus by accelerating immunity to pathogens[30–33].

T cell immunity plays a pivotal role in the control of coronavirus infections[34]. Yet, coronavirus infections induce a typical delayed type-I interferon (IFN) response, which has been shown to associate with impaired T cell response and worsened disease outcome[7–9,35]. Restoring the early type-I interferon response reduces virus-induced pathology[7], while timely administration of interferon before the viral peak was shown to be effective in treating COVID-19 patients[36,37]. In line with these, IBIS was superior in activating virus-specific CD4 + T cells in the lungs and BAL of vaccinated mice when compared with its counterpart that does not carry the IFNβ expression cassette (Fig. 7). Type-I IFN acts on multiple cellular targets to regulate antiviral T cell response both directly and indirectly during acute infection[38–40]. For instance, type-I IFN was shown to directly act on antiviral CD4 + T cells to protect them from natural killer cell-mediated killing in an acute LCMV infection model[20]. It also promotes the maturation of dendritic cells and so the antigen presentation and T cell co-stimulation[41,42]. Regulatory T cell-mediated suppression of antiviral CD4 + T cells is also inhibited by type-I IFN[22,23]. The function of type-I interferon in promoting T cell survival and expansion may help explain, at least partially, the increased virus-specific CD4 + T cell activation following vaccination with IBIS.

In this report, we generated a version of IBIS that expresses mouse IFNβ for mouse and hamster vaccination as a proof of concept. To help translate the findings to human application, a human version of IBIS (hIBIS) that expresses human IFNβ has been generated. Initially, an attempt was made to rescue the hIBIS in Vero A9 cells but in vain. We

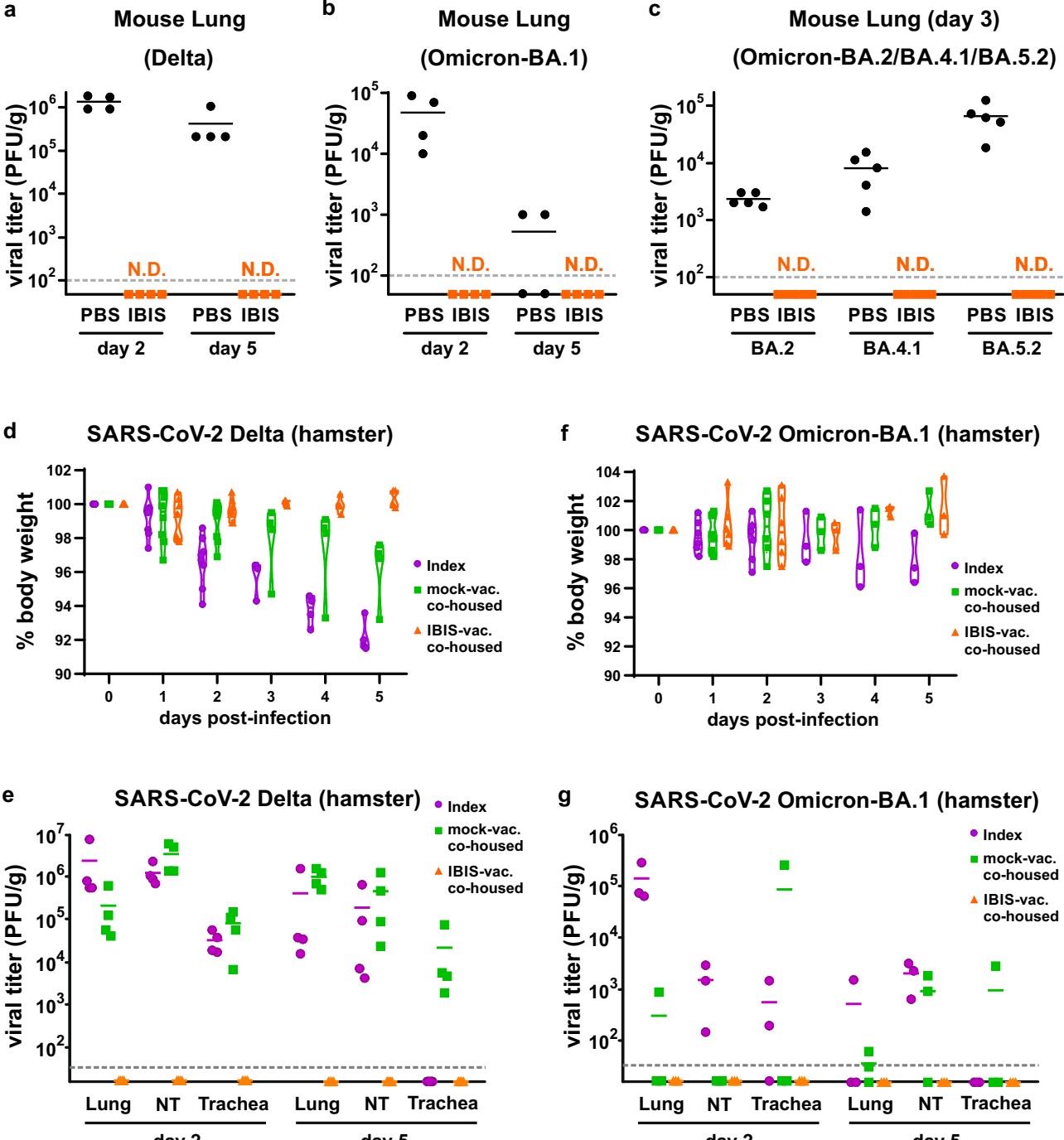

**Fig. 5 | IBIS vaccination prevents Delta and Omicron variants infection and transmission. a–c** Mouse infection. Six to ten week-old male K18-hACE2 transgenic mice were either mock-vaccinated or vaccinated with 2 doses of IBIS vaccine (1×10^6 PFU/dose) as described in Fig. 3a. At 14 days post second-dose of vaccination, the mice were infected intranasally with either SARS-CoV-2 Delta variant (2×10^3 PFU) (**a**) or Omicron-BA.1 variant (1×10^4 PFU) (**b**). Lungs of infected mice were harvested at day 2 and day 5 post-infection to determine the lung viral load by plaque assay; n = 4 per group. Similarly, PBS- or IBIS-vaccinated mice were infected with either Omicron-BA.2, BA.4.1 or BA.5.2 variants (1×10^4 PFU), and the lungs were harvested at day 3 post-infection for lung viral load determination (**c**); n = 5 per group. **d–g** Hamster infection and transmission. Six to ten week-old female hamsters were either mock-vaccinated or vaccinated with 2 doses of IBIS vaccine (3×10^6 PFU/dose) as described in Fig. 4a. On day -1, index hamsters were infected intranasally with either SARS-CoV-2 Delta variant (2×10^3 PFU) or Omicron-BA.1 variant (1×10^4 PFU). On day 0, the infected index hamsters were co-housed with mock- and IBIS-vaccinated hamsters, and then separated at 24 h after co-housing. Body weight of the hamsters was monitored for 5 days post-infection/co-housing (**d, f**). Lungs, nasal turbinate (NT) and trachea of the hamsters were harvested at day 2 and day 5 post-infection/co-housing for the determination of viral load by plaque assay (**e, g**). Delta infection and transmission experiment, n = 4 per group; Omicron-BA.1 infection and transmission experiment, n = 3 per group. Horizontal dotted-lines indicate detection limits for viral load determination by plaque assays. N.D., not detected. Source data are provided as a Source Data file.

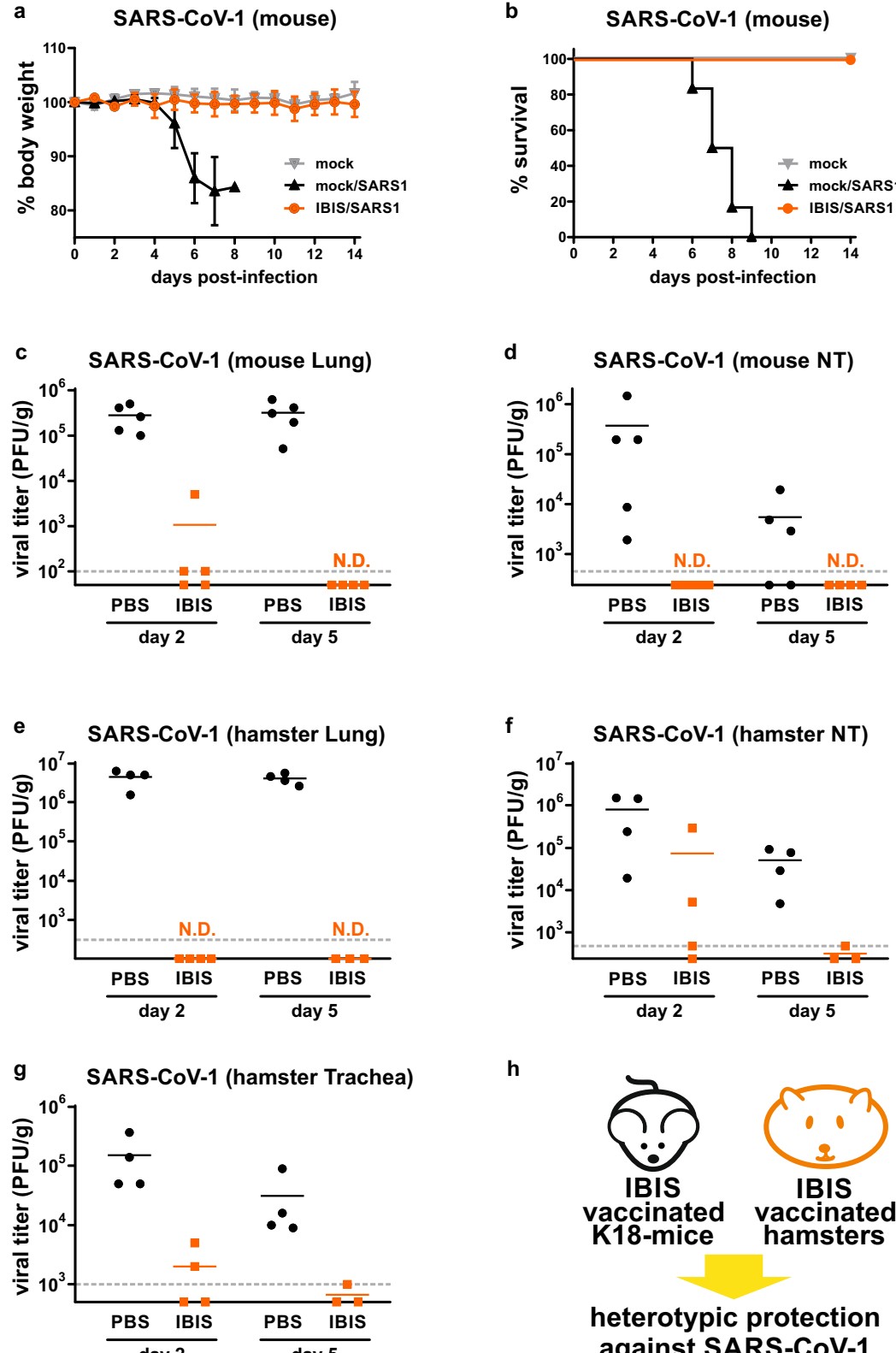

suspected that VeroE6 which is of monkey origin could be highly sensitive to human IFNβ. We therefore sought to abolish type-I interferon signaling in Vero A9 cells by knocking out STAT1 and generated the Vero A9B21 cells. Using the interferon-insensitive Vero A9B21 cells, hIBIS could be successfully rescued and grow to a titer comparable to mouse IBIS. Vero A9B21 was therefore used for the later production of both mouse and human IBIS. This observation also hinted that IFNβ

expression by IBIS can function as a safeguard mechanism to prevent IBIS from regaining the ability of productive infection. The globally circulating SARS-CoV-2 viruses are now predominated by Omicron variants, which display higher transmissibility, but lower pathogenicity and lethality compared to older strains. A second-generation human IBIS in Omicron backbone might further reduce the safety concerns about the use of attenuated SARS-CoV-2 viruses as live vaccines.

**Fig. 6 | IBIS vaccination confers heterotypic immunity against SARS-CoV-1 infection. a–d** Mouse infection. Six to ten week-old male K18-hACE2 transgenic mice were either mock-vaccinated or vaccinated with 2 doses of IBIS vaccine ($1×10^6$ PFU/dose) as described in Fig. 3a. At 14 days post second-dose of vaccination, the mice were infected intranasally with SARS-CoV-1 (GZ/50) ($2×10^3$ PFU). Body weight change (**a**) and survival (**b**) of the infected mice were monitored for 14 days post-infection. The percentage of body weight change is presented as mean ± SD. Mock-infected, n = 3; PBS-vaccinated/SARS-CoV-1 infected, n = 6; IBIS-vaccinated/SARS-CoV-1 infected, n = 6. Lungs (**c**) and nasal turbinate (**d**) of the infected mice were harvested at day 2 and day 5 post-infection for viral titer quantitation by plaque assay. PBS-day 2 (n = 5). IBIS-day 2 (n = 5). PBS-day 5 (n = 5). IBIS-day 5 (n = 4). **e–g** Hamster infection. Six to ten week-old female hamsters were either mock-vaccinated or vaccinated with 2 doses of IBIS vaccine ($3×10^6$ PFU/dose) at an interval of 14 days. At 14 days post second-dose of vaccination, hamsters were intranasally inoculated with SARS-CoV-1 (GZ/50) ($2×10^3$ PFU). Lungs (**e**), nasal turbinate (**f**) and trachea (**g**) were harvested at day 2 and day 5 post-infection for viral titer quantitation. PBS-day 2 (n = 4). IBIS-day 2 (n = 4). PBS-day 5 (n = 4). IBIS-day 5 (n = 3). Horizontal dotted-lines indicate detection limits for viral load determination by plaque assays. N.D., not detected. **h** Summary on IBIS-induced heterotypic protection against SARS-CoV-1 in mice and hamsters. Source data are provided as a Source Data file.

To this end, we have successfully generated two versions of IBIS that express mouse or human IFNβ. The virus-driven expression of IFNβ features three important characteristics. First, it activates innate antiviral response, acting as a second layer of self-limiting safeguard of the live vaccine in addition to envelope deletion. The antiviral IFNβ also suppresses co-infection and thus minimizes the chance of generating recombinant virus. Importantly, the replacement of orf8 with IFNβ improves the vaccine-elicited mucosal immunity by optimally activating antigen-specific CD4 + T cells in mice. As a result, IBIS vaccination not only prevented lethal infection and transmission of ancestral SARS-CoV-2 and the recent Delta and Omicron variants, but also was efficacious against the related sarbecovirus SARS-CoV-1 in two rodent animal models. Whether IBIS could be developed into the next-generation pan-sarbecovirus mucosal vaccine warrants further clinical trial studies.

## Methods

### Cell culture and virus-producing cell-lines
VeroE6, L929 and BHK21 cells were purchased from ATCC. A549-hACE2-hTMPRSS2 cells were purchased from Invivogen. VeroE6-hTMPRSS2 cells were purchased from Japanese Cancer Research Resources Bank. VeroE6 and A549-hACE2-hTMPRSS2 cells were cultured in Dulbecco's modified Eagle's medium (DMEM; Gibco). VeroE6-hTMPRSS2 cells were cultured in DMEM supplemented with 1 mg/mL geneticin (Gibco). L929 and BHK21 cells were cultured in Minimum Essential Medium (MEM; Gibco). All cultures were supplemented with 10% fetal bovine serum (FBS; Gibco) and cells were maintained in a humidified 37 °C incubator with 5% $CO_2$. VeroE6-eE and BHK21-eE cells were generated by transducing parental VeroE6 and BHK21 cells with lentivirus carrying an engineered SARS-CoV-2 E (eE) transgene followed by puromycin selection. VeroE6-eE cells were further single-cell sorted by fluorescence-activated cell sorting (FACS) and expanded. Transgene expression was verified by RT-qPCR. The VeroE6-eE cell clone A9 was further knocked out of the STAT1 gene using CRISPR-Cas9, and single-cell sorted by FACS to obtain the VeroE6-eE STAT1-KO clone A9B21 (Vero A9B21). L929-hACE2 cells were generated in-house by stable transduction with lentivirus encoding a human ACE2 transgene, followed by puromycin selection.

### Viruses
Ancestral SARS-CoV-2 HKU-001a (GenBank: MT230904), B.1.617.2/Delta variant (GenBank: OM212471) B.1.1.529/Omicron variants BA.1 (GenBank: OM212472), BA.2 (GISAID: EPI_ISL_9845731), BA.4.1 (GISAID: EPI_ISL_13777657), BA.5.2 (GISAID: EPI_ISL_13777658) and SARS-CoV-1 GZ50 (GenBank: AY304495) have been used in this study. All viruses were propagated in VeroE6 or VeroE6-hTMPRSS2 cells. Viral titer was determined by plaque assay in either VeroE6 or VeroE6-hTMPRSS2 cells and represented in plaque-forming unit per mL (PFU/mL). Cell-lines were inoculated at indicated multiplicity of infection (MOI) for 1.5 h in culture media containing 1% FBS, after which the inoculum was removed, and cells were replenished with fresh culture media containing 1% FBS. All infection experiments involving live SARS-CoV-1 and SARS-CoV-2 viruses were performed in Biosafety Level 3 (BSL3)

laboratory at The University of Hong Kong in accordance with the approved standard operation procedures.

### Recombinant virus and IBIS generation
Bacterial artificial chromosome (BAC) of wildtype SARS-CoV-2 HKU-001a was previously generated. BAC of SARS-CoV-2 virus with modified Envelope (SARS2-mE) were generated by introducing three premature stop codons into the Envelope gene using two-step red recombination[43]. IBIS for mouse and hamster experiments were generated by further introducing a mouse interferon-beta (mIFNβ) gene into the SARS2-mE BAC at the locus indicated in Fig. 1a. To rescue the SARS2-mE and IBIS virus, the BACs were transfected with Lipofecta-mine 2000 (Invitrogen) into BHK21-eE cells. Transfected BHK21-eE cells were trypsin-dissociated and co-cultured with Vero A9 or Vero A9B21 cells at 6–8 h post-transfection. Recombinant viruses generated were plaque-purified, further propagated, and quantitated by plaque assay in Vero A9 or Vero A9B21 stable cells. The recombinant viruses were then concentrated by ultra-centrifugation at 28000 rpm ($133900 × g$), 4 °C for 4 h against a 25% sucrose bed on Optima XPN-100 ultracentrifuge with SW32Ti adaptor (Beckman Coulter). Absence of replicative virus in the SARS2-mE and IBIS virus stock was confirmed by plaque assay using parental VeroE6 cells, which showed complete absence of plaques and cytopathic effect (CPE). IBIS containing human IFNβ was produced in Vero A9B21 cells exclusively.

### Animals
K18-hACE2 transgenic mice (2B6.Cg-Tg(K18-ACE2)2Prlmn/J) were originally obtained from The Jackson Laboratory. Golden Syrian hamsters (strain HsdHan®:AURA) were originally sourced from Envigo, USA. Animals were housed and bred under an AAALAC International accredited program at the Center for Comparative Medicine Research (CCMR), The University of Hong Kong. Animals were housed in open cages or individually ventilated cages under a 12:12 dark/light cycle within environmentally controlled rooms. Standard pellet feed and water were provided ad libitum. All animal experiments were performed with prior approval from the Committee on the Use of Live Animals in Teaching and Research (CULATR), The University of Hong Kong and under licence from the Hong Kong SAR Government's Department of Health.

### Animal vaccination, infection and tissue harvest
Sex- and age-matched animals were randomized into experimental groups. Six to ten week-old animals were anesthetized with an intra-peritoneal injection of ketamine and xylazine prior to vaccination or infection. For IBIS vaccination, $1×10^6$ PFU/mouse in 20 μL PBS, or $3×10^6$ PFU/hamster in 50 μL PBS was intranasally inoculated into the nostril of each anesthetized animal if not specified. For mRNA vaccination, one microgram of BioNTech BNT162b2 mRNA vaccine (Pfizer) was intra-muscularly injected into the hind-limb muscle of mice. Fourteen days after vaccination, a booster of the same strength was given in the route same as the first vaccination. Blood was collected from the facial vein of mice or the gingival vein of hamsters under anesthesia. Animals were infected by intranasal inoculation of SARS-CoV-1 or SARS-CoV-2

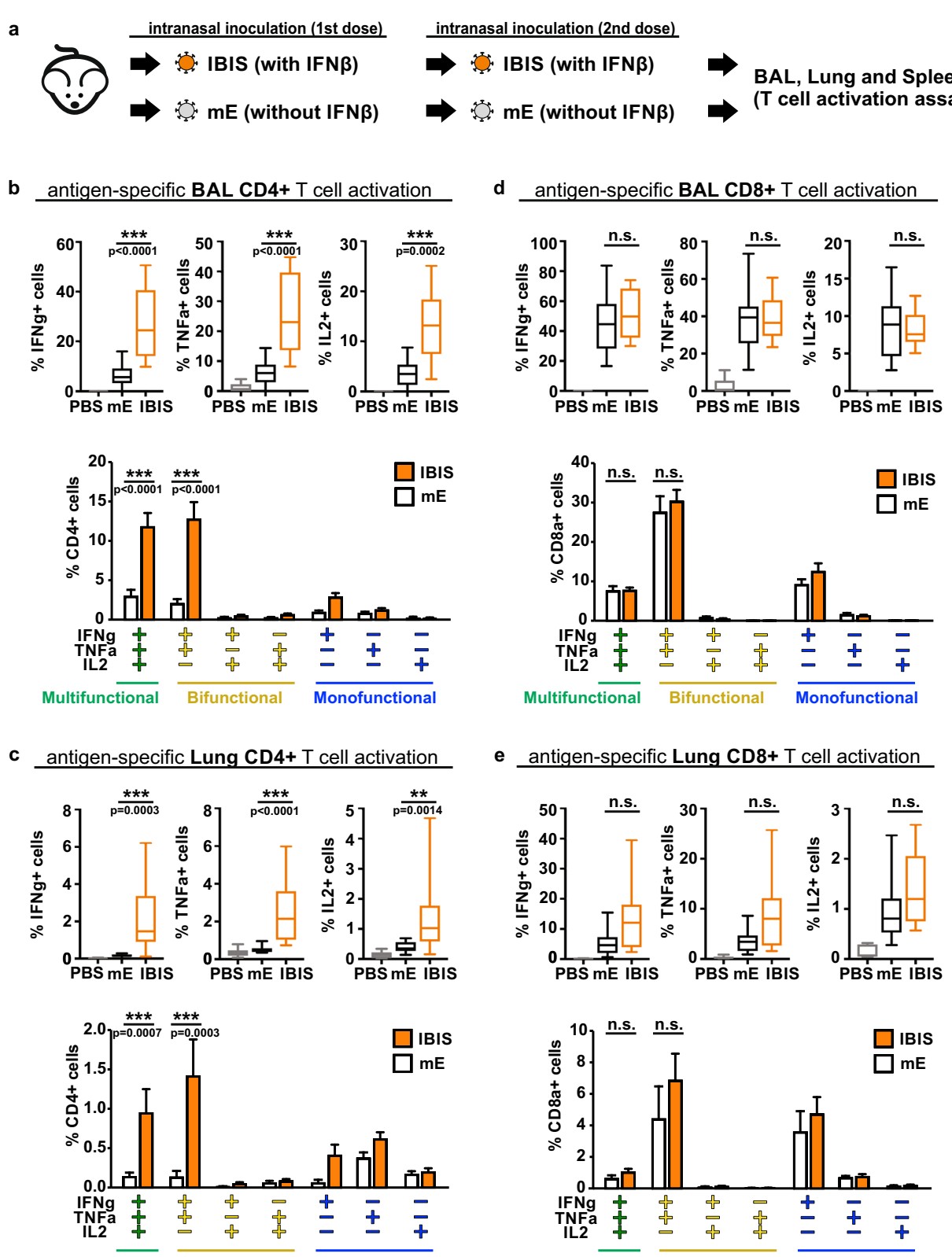

virus diluted in 20 μL (mouse) or 50 μL (hamster) PBS under anesthesia. All animal experiments involving BSL3 agents were performed in Biosafety Level 3 Animal Facility at The University of Hong Kong in accordance with the approved standard operation procedures and CULATR protocols. Infected animals were housed in individually ventilated cages. Body weight and disease symptoms of infected animals were monitored daily for 14 days. At experimental endpoints or

humane endpoints, animals were sacrificed by intraperitoneal injection of sodium pentobarbital, except for assays involving bronchoalveolar lavage (BAL) fluid collection in which mice were sacrificed by overdose of isoflurane. Tissues harvested were homogenized in cold PBS using TissueRuptor II (Qiagen) or with plastic pestles. Homogenate was then centrifuged at 3220 g, 4 °C for 10 min. The clear supernatant was aliquoted and stored at −80 °C for future assays. For

**Fig. 7 | Interferon integration in IBIS enhances mucosal antigen-specific multifunctional CD4 + T cell activation. a** Schematic diagram showing the experimental setup for T cell activation assay. Six to ten week-old female K18-hACE2 transgenic mice were either mock-vaccinated, or vaccinated with two doses of IBIS (5×10⁴ PFU) or SARS2-mE (5×10⁴ PFU) at an interval of 14 days. At 7 days post second-dose of vaccination, mice were sacrificed for T cell assay. Transcardiac perfusion was performed after euthanasia, bronchoalveolar lavage (BAL) and lung were then harvested for the collection of immune cells. Cells isolated from BAL and dissociated lung tissues were evaluated by T cell activation assay. The isolated cells were stimulated by SARS-CoV-2 spike glycoprotein peptide pool (ancestral strain), followed by immunofluorescence staining for surface markers (CD3, CD4 and CD8) and intracellular cytokines (IFNγ, TNFα and IL2). Flow cytometry analysis was performed to determine the activation of antigen-specific CD4 + T cells in BAL (**b**) and dissociated lungs (**c**); and CD8 + T cells in BAL (**d**) and dissociated lungs (**e**). Activation of antigen-specific CD4 + T cells or CD8 + T cells was presented as a percentage of the total cell count of CD4 + T cells or CD8 + T cells respectively (upper panels). Center line, median; box limits, upper and lower quartiles; whiskers, minima and maxima. Bar graphs show the percentage of multifunctional, bifunctional and monofunctional antigen-specific CD4 + T cells or CD8 + T cells (lower panels) and data are presented as mean ± SEM. The results are from 4 independent experiments with the total number of mice per group: PBS (n = 10). mE (n = 11). IBIS (n = 13). mE, SARS2-mE. Statistical analysis was performed using two-sided non-parametric Mann-Whitney test. ***, p < 0.001; **, p < 0.01; n.s., not significant. Source data are provided as a Source Data file.

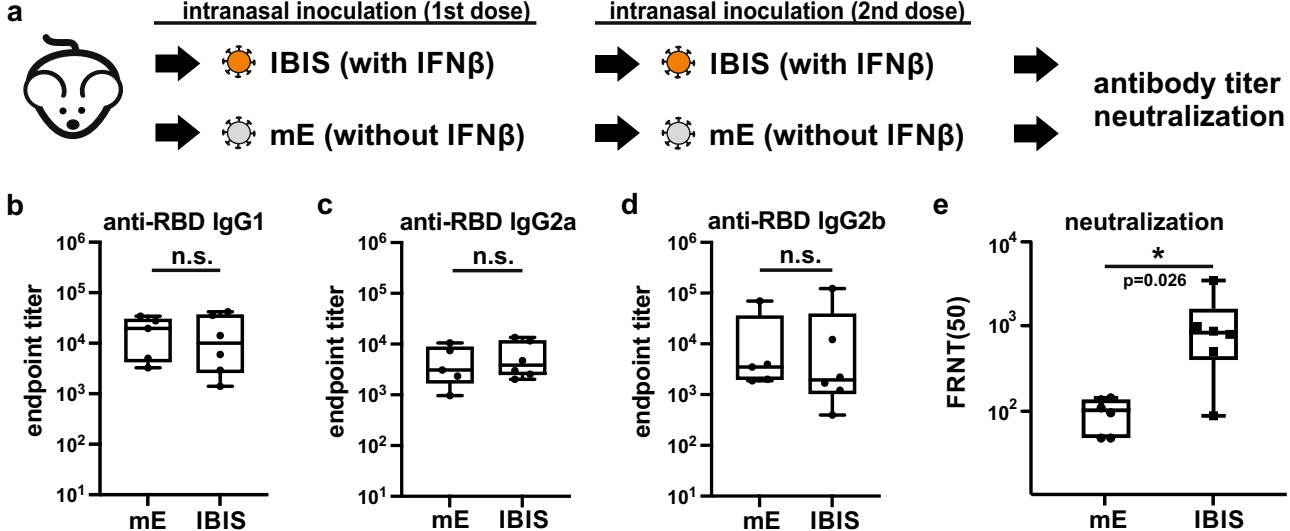

**Fig. 8 | Interferon integration in IBIS slightly enhances neutralization activity against authentic SARS-CoV-2 virus. a** Schematic diagram showing the experimental setup for antibody response comparison. Six to ten week-old male K18-hACE2 transgenic mice were vaccinated with two doses of either IBIS (3×10³ PFU/dose) or SARS2-mE (mE) (3×10³ PFU/dose) at an interval of 14 days. At 14 days post second-dose of vaccination, sera of vaccinated mice were harvested to determine the titer of RBD (ancestral)-specific IgG1 (**b**), IgG2a (**c**), and IgG2b (**d**) antibodies. Neutralization titer of the sera was determined by FRNT₅₀ against authentic SARS-CoV-2 (ancestral) virus (**e**); SARS2-mE, n = 5; IBIS, n = 6. Center line, median; box limits, upper and lower quartiles; whiskers, minima and maxima. Two-sided non-parametric Mann-Whitney test was performed to calculate the statistical significance. *, p < 0.05; n.s., not significant. Source data are provided as a Source Data file.

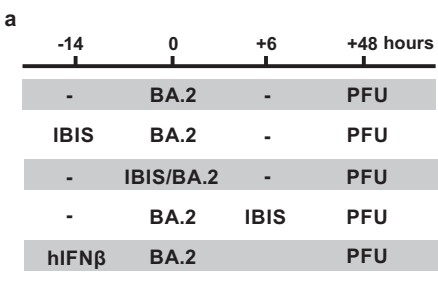

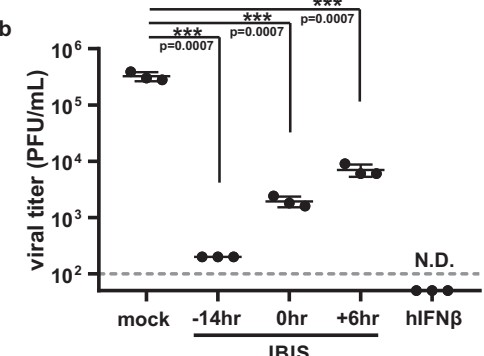

**Fig. 9 | IBIS encoding human IFNβ suppresses SARS-CoV-2 co-infection in human cells.** A human version of IBIS (hIBIS) encoding human IFNβ was generated. **a** Schematic timeline of the co-infection experiment. Human lung fibroblasts A549-hACE2-hTMPRSS2 cells were infected with SARS-CoV-2 (Omicron-BA.2) at MOI = 0.1, together with pre-, co- or post-treatment of hIBIS at MOI = 0.2. A group with recombinant human IFNβ (1000 IU/mL) pre-treatment was included as a positive control for the inhibition. At 48 h post-infection, culture supernatant was collected for viral titer quantitation by plaque assay (**b**); n = 3 biological replicates per group. Horizontal dotted-line indicates the detection limit for viral load determination by plaque assay. Data are presented as mean ± SD. Statistically analysis was performed using two-sided unpaired student t test. ***, p < 0.001; N.D., not detected. Source data are provided as a Source Data file.

histology, tissues were fixed in 4% paraformaldehyde (PFA) in PBS for >24 h, followed by paraffin embedding, sectioning, and hematoxylin and eosin (H&E) staining or immunofluorescence (IF) staining.

## VSV-GFP interferon bioassay

Functional mouse IFNβ protein was detected by interferon bioassay. Briefly, samples were diluted in MEM with 10% FBS according to the indicated dilution. One milliliter of diluted sample was treated onto L929 cells in a 12-well plate. At 24 h post-treatment, the inoculum was removed and the L929 cells were further infected with vesicular stomatitis virus containing a GFP reporter (VSV-GFP). At 24 h post-infection, infected cells were fixed with 4% PFA and GFP was observed using a fluorescence microscope. The absence of GFP indicates the presence of functional mouse IFNβ protein in a concentration high enough to protect treated cells from VSV-GFP infection.

## Enzyme-Linked Immunosorbent Assay (ELISA)

Mouse IFNβ and TNFα proteins were quantitated using mIFNβ and mTNFα Quantikine ELISA kits (R&D Systems) respectively according to the manufacturer's instructions. Anti-receptor-binding domain (RBD) and anti-nucleoprotein (N) antibodies were quantitated by in-house ELISA. Briefly, high-binding ELISA plates (Corning) were coated with recombinant RBD or N protein, casein-blocked and loaded with serially diluted animal sera. After washing, biotin-conjugated anti-mouse IgG or anti-hamster IgG antibodies (BioLegend) were added, followed by the addition of HRP-Streptavidin (BioLegend). After thorough washing, TMB-substrate (Thermo Fisher Scientific) was then added to allow color development and sulfuric acid for termination of the reaction. Absorbance at 450 nm was measured using Varioskan LUX multimode microplate reader (Thermo Fisher Scientific). A cut-off equal to the baseline OD value plus ten times of its standard deviation was set. The serum dilution at which the OD value dropped to this cut-off was calculated using GraphPad Prism software (v.8.0) to represent the endpoint titer.

## Hematoxylin and Eosin (H&E) staining and immunofluorescence (IF) staining

Animal tissues were fixed in 4% PFA for >24 h and paraffin-embedded. Tissues were then sectioned and mounted onto microscope slides, followed by de-waxing in xylene, and staining with hematoxylin and eosin solution. Stained slides were then scanned by Cytation 7 cell imaging multi-mode reader (BioTek). For IF staining of cultured cells, cells were infected at the indicated MOI. Infected cells were fixed with 4% PFA for 30 min, NP-40 permeabilized and blocked with 5% normal donkey serum (Jackson ImmunoResearch). Nucleoprotein of SARS-CoV-2 was then stained with rabbit anti-SARS-CoV-2 N antibody (in-house), followed by Alexa Fluor 488-conjugated donkey anti-rabbit antibody (Abcam). Fluorescence was then detected using a fluorescence microscope. For IF staining of animal tissue sections, antigen retrieval was performed by microwave heating the slides in citrate-based antigen unmasking solution (Vector Laboratories), followed by blocking in 5% normal goat serum (Gibco). The slides were then stained with anti-mouse B220-APC antibody (RA3-6B2; BioLegend). Alternatively, slides were stained with rabbit anti-SARS-CoV-2 N antibody (in-house) or rabbit anti-CD4 (EPR19514; Abcam) followed by Alexa-Fluor-488 conjugated anti-rabbit antibody (Thermo Fisher Scientific). Nuclei of cells were counter-stained with Hoechst 33258 (Thermo Fisher Scientific) and autofluorescence was quenched by Sudan Black B. Fluorescence images were captured with LSM900 confocal microscope (Zeiss).

## Focus reduction neutralization test (FRNT)

Focus reduction neutralization test (FRNT) was performed to quantitate serum-neutralizing antibodies against authentic SARS-CoV-2 virus. Briefly, animal sera were serially diluted. The diluted serum was then mixed with an equal volume of 300 focus-forming units (FFU) of SARS-CoV-2 virus. The serum/virus mix was incubated at 37 °C for 1 h, and then transferred to VeroE6 cells seeded in 96-well plates. After 1-h adsorption, the inoculum was aspirated. Cells were then PBS-washed and replenished with fresh DMEM containing 1% FBS. At 6 h post-infection, cells were PFA-fixed for immunostaining against SARS-CoV-2 N protein. Stained plates were scanned using Cytation 7 cell imaging multi-mode reader (BioTek) and the number of foci in each well was counted with Gen5 software (BioTek). The serum dilution at which half number of foci could be observed was calculated to represent the 50% reduction in $FRNT_{50}$ or 75% reduction in $FRNT_{75}$ using GraphPad Prism software (v.8.0).

## RNA extraction and Reverse Transcription-quantitative PCR

Cultured cells or animal tissues were lysed in RNAiso Plus reagent (Takara). RNA was extracted according to the manufacturer's instructions. RNA extracted was then reverse-transcribed into cDNA using PrimeScript™ RT reagent Kit with gDNA Eraser (Takara). Transcript levels of viral RNA and 18 S ribosomal RNA were determined by quantitative PCR (qPCR) on ViiA 7 Real-Time PCR System (Thermo Fisher Scientific), using SYBR Premix Ex Taq reagent (Takara). The viral transcript was normalized to the transcript level of 18 S rRNA and presented as a relative expression. The following primers were used: Orf1ab-F: 5′-GCTTG ATGGC TTTAT GGGTA G-3′. Orf1ab-R: 5′-TGAAT TGTGA CATGC TGGAC-3′. 18S-F: 5′-GTAAC CCGTT GAACC CCATT-3′. 18S-R: 5′-CCATC CAATC GGTAG TAGCG-3. Alternatively, the transcript level of the RNA was determined by TaqMan probe-based one-step RT-qPCR using QuantiNova Probe RT-PCR Kit (Qiagen) and expressed as a number of viral RNA copies per microgram of RNA input. The following primers and probe were used: Forward primer: 5′-CGCAT ACAGT CTTRC AGGCT-3′. Reverse primer: 5′-GTGTG ATGTT GAWAT GACAT GGTC-3′. FAM-probe: 5′-TTAAG ATGTG GTGCT TGCAT ACGTA GAC-3′.

## Multiplex cytokine analysis

Cytokine levels in bronchoalveolar lavage (BAL) fluid obtained from vaccinated mice were analyzed by bead-based multiplex assay (LEGENDplex™; BioLegend), according to manufacturer's instruction. Samples were acquired on BD LSRFortessa Cell Analyzer and data were analyzed by LEGENDplex™ Cloud-based Data Analysis Software (BioLegend).

## Sample preparation and library construction for Single-cell RNA sequencing

Vaccinated mice were sacrificed by overdose of isoflurane at 18 h post-vaccination. Bronchoalveolar lavage (BAL) was collected by flushing the lungs with a total of 2 mL PBS through trachea using a 22 G catheter. BAL cells were then resuspended in PBS/5% FBS for downstream processing. Library construction and sequencing were performed at the Center for PanorOmic Sciences, HKU using the 10x Genomics platform. Briefly, single-cell encapsulation and cDNA libraries were prepared by Chromium Next GEM Single Cell 5′ Reagent kit v2 (Dual Index) and Chromium Next GEM Chip K Single Cell kit according to the manufacturer's instructions. Input cell number was then normalized to 22,750 cells across samples. Reverse transcription, cDNA cleanup and amplification were performed on Gel Beads-in-emulsions. Library size distributions were determined by Agilent 2100 Bioanalyzer. Sequencing was performed using Illumina NovaSeq 6000 for Pair-End 151 bp sequencing, and sequencing reads were imported to 10X CellRanger pipeline for data preprocessing and initial filtered feature barcodes and matrices were exported.

## Single-cell RNA sequencing data analysis

Based on the filtered barcodes and matrices generated from 10X CellRanger, the sequencing data were further analyzed using Seurat v3 on R 4.3.0. Briefly, data was merged, filtered, normalized, scaled,

dimensionality reduced, and clustered. Merged data was distinguished based on the assigned original identity. Differential gene expression between clusters was determined by Wilcoxon rank sum test in Seurat. Cell type identity was validated by evaluating differential gene markers. UMAP and violin plots were generated and visualized using Seurat and ggplot2 functions.

## T cell activation analysis

Cell culture reagents were purchased from Gibco (USA), unless otherwise stated. Tissue isolation: Mock-vaccinated mice and mice vaccinated with two doses of IBIS or SARS2-mE were sacrificed at 7 days post second-dose of vaccination. Mice were euthanized by an overdose of isoflurane and transcardiac perfusion, which aimed to minimize the contamination of tissue resident cells by blood borne cells, was performed with 10 mL of 1% penicillin/streptomycin in PBS before tissue isolation. Bronchoalveolar lavage (BAL) and lung were harvested for the collection of immune cells. Lung tissues were minced and digested in RPMI 1640 containing 5% fetal bovine serum (FBS), 1 mg/mL collagenase D (w/v; Roche), 0.15 mg/mL DNase I (w/v; Roche) and 1% penicillin/streptomycin at 37 °C for 30 min. Digestion was halted by an addition of EDTA at a final concentration of 5 mM, followed by an incubation on ice for 15 min. Digested tissue pieces were then mashed through a 100-μm cell strainer. Resulting cell suspension was incubated in the lysing solution (BD Biosciences) for 1 min, followed by two washes with 1% penicillin/streptomycin in PBS to remove erythrocytes.

T cell activation and culture: Cells obtained from BAL or single cell suspension of lung tissues after red blood cell lysis were resuspended in RPMI 1640 containing 10% FBS, 2 mM L-Glutamine, 55 μM 2-Mercaptoethanol, 2 mM sodium pyruvate, 0.1 mM MEM non-essential amino acids and 1% penicillin/streptomycin. All cells obtained from BAL or 10% of total cells obtained from a dissociated lung were cultured (37 °C; 5% $CO_2$) in a 96-well round-bottom plate overnight in the presence of 1 μg/mL SARS-CoV-2 spike peptide pool, which contains 315 peptides (15mers with 11 amino acid overlap) of SARS-CoV-2 Spike glycoprotein (Swiss-Prot ID: P0DTC2; PepMix™; JPT). Cells treated with 50 ng/mL Phorbol 12-myristate 13-acetate (PMA; Sigma-Aldrich) plus 1 μg/mL Ionomycin (Sigma-Aldrich) or DMSO (Sigma-Aldrich) alone served as positive and negative controls respectively. Brefeldin A (Sigma-Aldrich) and Monensin (BioLegend), at a final concentration of 7 μg/mL and 2 μM respectively, were added to the culture after 2 h of incubation.

Immunofluorescence staining for flow cytometric analysis: Cultured cells were added with EDTA solution at a final concentration of 20 mM, incubated for 15 min at room temperature and then transferred to a 96-well v-bottom plate for staining. Cells were washed once with PBS, incubated with Fc-receptor blockade (anti-mouse CD16/32 mAb; 2.4G2; BD Biosciences) and stained with antibody cocktail containing anti-mouse CD3-SB780 (17A2; eBioscience), CD4-BB700 (RM4-5; BD Biosciences), CD8-APC/Fire750 (53-6.7; BioLegend) and Zombie Aqua Fixable Viability dye (BioLegend) in PBS for 15 min at 4 °C. Stained cells were washed once in staining buffer (5% FBS in PBS), fixed and permeabilized using Fixation/ Permeabilization Solution Kit (BD Biosciences) according to the manufacturer's instructions. To detect the presence of intracellular cytokines, cells were further stained with anti-mouse TNFα-BV421 (MP6-XT22), IL2-PE (JES6-5H4) and IFNγ-APC (XMG1.2) (all from BioLegend) at 4 °C for 1 h prior to wash and resuspension in staining buffer. Samples were acquired by BD LSRFortessa Cell Analyzer and data were analyzed by FlowJo software (BD Biosciences). Quadrant gates shown in fluorescence-activated cell sorting (FACS) plots were set according to fluorescence-minus-one (FMO) controls using corresponding isotype matched control antibodies. Statistical differences were evaluated by Mann-Whitney test.

## Statistical analysis

For quantitation of samples derived from cell cultures, statistical analysis was performed using unpaired parametric student t test. Unpaired non-parametric Mann-Whitney test was used for statistical analysis of animal data.

## Reporting summary

Further information on research design is available in the Nature Portfolio Reporting Summary linked to this article.

## Data availability

The scRNA-seq data generated in this study have been deposited in the NCBI Sequence Read Archive under the BioProject accession code PRJNA1002063. Source data are provided with this paper.

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

## Acknowledgements

This work was supported by General Research Fund 17123420; Health@InnoHK (funded by Innovation and Technology Commission, the Government of the Hong Kong Special Administrative Region, China); and Theme-based Research Scheme (T11-709/21-N). In addition, this study was partially supported by grants 2021YFC0866100 and 2023YFC3041600 from the National Key Research and Development Program of China; and emergency collaborative grant EKPG22-01 from Guangzhou Laboratory.

## Author contributions

C.-K.Y., L.-F.M., and W.-M.W. contributed equally to this study. K.-Y.Y. and K.-H.K. contributed equally to this study. C.-K.Y., L.-F.M., W.-M.W., J.-Y.L., L.-Y.C., D.-T.Y.C., Y.-Y.N., A.-C.Y.L. and K.-H.K. contributed to data acquisition. C.-K.Y., L.-F.M. and K.-H.K. contributed to data interpretation. K.H.K. contributed to the conception and design of experiments. K.-H.K. and C.-K.Y. have drafted the manuscript. K.-H.K., C.-K.Y., N.Z. and K.-Y.Y. have revised the manuscript. All authors have approved the submitted manuscript.

## Competing interests

The authors declare the following competing interests. The authors applied for US and PCT patents related to this work (Inventor: K.-H.K.).
