## [Peer Review File · Nature Communications]

Reviewers' Comments:

Reviewer #1:

Remarks to the Author:

Dear Editor,

The manuscript 'An interferon-integrated pan-sarbecovirus vaccine' by Yuen et al. provides a well-written novel insight into a potential mucosal vaccine against SARS-CoV-2 with pan-coronavirus protective potential. The experiments are well-designed, and I particularly like the vaccine design. This manuscript adds to the current insight within the SARS-CoV-2 field. I have a few comments, followed by minor comments, which I would like to see the authors address.

Comments:

- The authors clearly show the functionality of the mouse-IFN β in their vaccine in murine cells, but they do not do so in hamster cells or in vivo. However, the vaccine is used in both mice and hamsters. Could the authors prove the expression of mouse IFN β does make a difference in the mouse by e.g. looking at the expression of IFN β directly or the induction of ISGs, comparing the IBIS and mE?
- The authors very nicely show the expression of IFN β in the mouse lung (Figure 2E), but what about the induction of ISGs? In other words, is the IFN β functional in the mouse?
- Please describe any symptoms the mice had upon vaccination with IBIS (e.g. weight loss)
- Please provide all experimental details in the figure legends and in the text, including time between vaccination and challenge, number of animals used, dose, and route.
- I really like the cellular immunity done in Figure 7. However, it has to be acknowledged that one can not distinguish between circulating and tissue-resident T cells. I highly recommend the authors perform an intravenous CD45 antibody injection prior to euthanasia in future studies. It is also unclear whether a mock-vaccinated group was used, since this is mentioned in the legend, but is not present in the figure or in the methods. I do think this would have been an important control. Finally, I am missing the gating strategy. I assume the authors gated on CD45, CD3, before they looked at CD4 and CD8, but this is not mentioned in the text or the figure. Could you please clarify? Also, was a live/dead stain done?
- Can you please comment on what you think would happen if you would vaccinate someone with your vaccine who has an active virus replication? I understand Figure 9 shows that if you superinfect early on, it will abrogate infection, but what will happen at a later stage? Will this be a concern?
- Why does the time between the last vaccination and infection change between 14 and 28 days?
- Would it be possible to get the lung histology scored?
- Please explain what the dark discoloring is in the lungs in Figure S4, IBIS-vac/SARS2.

Minor comments:

- Please provide stats paragraph in methods. Please note that one cannot use a student's T test in animal studies with smaller numbers. A Mann-Whitney test is more appropriate. I do not think it will change the significance of your studies.
- Ensure all abbreviations are in the figure legends (e.g. S/N), and all stat abbreviations *, **, ***
- Ensure spelling is consistent throughout (e.g. delta vs Delta, n=6 vs N=6)
- Please make it clear throughout that IFN β replaced the orf8. this is e.g. not clear in sentence 109 and creates confusion.
- Please explain why the 90 synonymous changes in the eE transgene in the stable cell line will prevent the recombination (line 138)
- Please specify for all figures and in the text which SARS-CoV-2 variant you use
- Legend Figure 1B; it looks like it can still replicate in cells it infected, just not multiple rounds.
- Supplementary Figure S2 is very hard to read.
- Is there a way to quantify the fluorescence?
- Please compare VN titers in Figure 3B to the BNT-vaccinated animals.
- Line 216: Please explain why IgM plays a particular role in virus neutralization, and do you expect to see this long-term?
- Experiment Figure 3, please explain why mRNA mice were not challenged also.
- Legend Figure 3A: missing description mRNA mice
- Figure 4A: states day -29 and -15 in figure, but day -28 and -14 in legend. Please clarify.
- Line 227: I do not believe you can claim sterilizing immunity; you just have not been able to

detect replicating virus. This does not mean that the virus did not replicate somewhere.

- Please split up Figures 5E and G so they are easier to read.
- The authors do not discuss the pie charts in Figure 7, so I suggest you either remove them or you perform stats on them and discuss them in the text.
- Figure 8, specify against what antigen (ancestral?)
- Figure 9, please show in log scale, like all others are. Also, the legend states there is a grey dotted line, but I do not see it.
- I do not think panels S9G and S10G are needed
- Please remove panel S11A, since it is also in 7A

Reviewer #2:

None

Reviewer #3:

Remarks to the Author:

This manuscript by Yuen et al., reported the construction of a SARS-CoV-2 virus that is envelope-deficient and has the orf8 replaced by interferon-beta. Such an interferon-Beta-Integrated SARS-CoV-2 (IBIS) was subsequently tested for its potential as a live attenuated vaccine in K-18 hACE2 mice and in Syrian hamsters. While the authors presented some interesting results, I have the following comments for the authors to address:

1) The authors described IBIS as single-round replicative (line 150, line 160, Fig 1 and S2, etc). Whether IBIS is attenuated because of defect in virus replication or assembly should be carefully verified. In order for immunostaining to work (Fig. 1), the virus has to be replication competent. Otherwise, no viral antigens would be available for staining. If the attenuation is caused by defect in assembly, perhaps electron microscopy or CryoEM can shed light on that. Of note, SARS-CoV that lacks E is attenuated (JVI 2007, p. 1701; JVI 2008, p. 7721).

2) IBIS appears to be very immunogenic in animal models, which is a bit counterintuitive. If IBIS infection is only single-round, how would vaccination induce measurable amount IFN- β in mouse lungs? The virus would not even be able to arrive at lungs to express IFN- β . Can authors show how responsive IBIS is to IFN β treatment in comparison to wild type virus? Is IFN- β in IBIS an immunogen instead?

3) Similarly, IBIS appeared to offer protection that has not even be achieved in natural infection in that it induces sterile immunity (line 41 and 227). Do the authors know the kinetics of IBIS after vaccination? Is the vaccine virus only detectable in nasal cavity? What is the mechanism for its induction of protection in the lungs if the vaccine virus only infects the nasal cavity? In multiple figures, the authors showed viral titers from IBIS-vaccinated animals at levels below detection limit. How was the viral titer determined? By plaque assay or by quantifying the immunostained lung sections?

Point-to-point address

Reviewer #1:

Comment #1:

The manuscript 'An interferon-integrated pan-sarbecovirus vaccine' by Yuen et al. provides a well-written novel insight into a potential mucosal vaccine against SARS-CoV-2 with pan-coronavirus protective potential. The experiments are well-designed, and I particularly like the vaccine design. This manuscript adds to the current insight within the SARS-CoV-2 field. I have a few comments, followed by minor comments, which I would like to see the authors address.

Address #1:

Thanks for all valuable comments and we have done additional experiments to improve the manuscript accordingly.

Major comments (#2-#10)

Comment #2:

- The authors clearly show the functionality of the mouse-IFN β in their vaccine in murine cells, but they do not do so in hamster cells or in vivo. However, the vaccine is used in both mice and hamsters. Could the authors prove the expression of mouse IFN β does make a difference in the mouse by e.g. looking at the expression of IFN β directly or the induction of ISGs, comparing the IBIS and mE?

Address #2:

Thank you for the comment again. To confirm the specific production of IFN β by IBIS in vivo and its function, additional experiments (cytokine profile and scRNA-sequencing) were performed comparing IBIS and mE as suggested and these new data were added in Supplementary Figure 3. Briefly, K18-hACE2-tg mice were intranasally vaccinated with either IBIS or its counterpart that does not contain the IFN β cassette (mE). Eighteen hours later, bronchoalveolar lavage (BAL) was collected. BAL fluid was subjected to cytokine quantitation, while the cells were subjected to single-cell RNA sequencing.

Comparing IBIS to mE, elevated amount of IFN β , together with IP-10 which is an interferon-induced chemokine, could be detected in the BAL fluid (Supplementary Figure 3b-3c). Single-cell analysis also indicated that subsets of immune cells displaying ISG signature were induced specifically by IBIS compared to mE (Supplementary Figure 3d-3f).

The data strengthened the conclusion that IBIS produced functional IFN β that can induce ISGs in vivo. Description of these data were added in the text (Line:197-201).

Note: Prior to testing the IBIS vaccine in hamsters, we performed a pilot experiment to confirm the mouse IFN β is functional in hamster, which is another rodent susceptible to SARS-CoV-2 and SARS-CoV-1 infection. 1×10^5 IU of recombinant mouse IFN β was intranasally instilled into a hamster, and then the hamster lungs were harvested at 8 hours post-treatment. The hamster ISG induction was then confirmed by RT-qPCR.

Comment #3:

- The authors very nicely show the expression of IFN β in the mouse lung (Figure 2E), but what about the induction of ISGs? In other words, is the IFN β functional in the mouse?

Address #3:

As discussed in the response to comment #2, additional experiments were performed to confirm that ISGs were specifically induced in vivo by IBIS but not mE (Supplementary Figure 3), indicating that the IFN β produced was functional in vivo.

Comment #4:

- Please describe any symptoms the mice had upon vaccination with IBIS (e.g. weight loss)

Address #4:

We have added the body weight record, survival and disease scoring of vaccinated mice from day 0 to day 14 after vaccination in Supplementary Figure 4b-4d (Line: 204-205).

The vaccinated mice were apparently healthy. No disease-like symptoms could be observed in the vaccinated mice. No body weight change, nor disease signs like ruffled fur, hunched back or laboured breathing were seen (Supplementary Figure 4b-4d). Animals remained active and responsive to external stimuli.

This is in line with the observation that no infectious virus could be detected in the mouse lungs at 2 days after IBIS intranasal vaccination (Figure 3g, right column).

Comment #5:

- Please provide all experimental details in the figure legends and in the text, including time between vaccination and challenge, number of animals used, dose, and route.

Address #5:

Thanks for the recommendation. All figure legends have been updated with detailed description of the experimental set-up accordingly.

Comment #6:

- I really like the cellular immunity done in Figure 7. However, it has to be acknowledged that one cannot distinguish between circulating and tissue-resident T cells. I highly recommend the authors perform an intravenous CD45 antibody injection prior to euthanasia in future studies. It is also unclear whether a mock-vaccinated group was used, since this is mentioned in the legend, but is not present in the figure or in the methods. I do think this would have been an important control. Finally, I am missing the gating strategy. I assume the authors gated on CD45, CD3, before they looked at CD4 and CD8, but this is not mentioned in the text or the figure. Could you please clarify? Also, was a live/dead stain done?

Address #6:

Thanks for the comment and advice. For all T-cell assays described in this manuscript, transcardiac perfusion with excess amount of PBS has been performed prior to BAL and lung harvest to minimize the contamination of tissue resident cells by circulating blood cells. We have now updated the figure legend and the method section to include this information together with other experimental details.

In addition, a PBS-vaccinated control group was included when performing the experiments and corresponding data have been added to Figure 7.

And yes, live/dead stain has been performed. Live cells were gated on CD3, and then CD4 and CD8. The gating strategy has now been included in the Supplementary Figure 12.

Thanks again for the useful advice and we will include the intravenous CD45 antibody in the future study.

Comment #7:

- Can you please comment on what you think would happen if you would vaccinate someone with your vaccine who has an active virus replication? I understand Figure 9 shows that if you superinfect early on, it will abrogate infection, but what will happen at a later stage? Will this be a concern?

Address #7:

This is a particular important concern to live vaccines, especially there could be vaccinees who are infected with SARS-CoV-2 but remain asymptomatic. At later stage, the potential viral recombination is one of the safety concerns. The best improvement of safety would be making a second-generation IBIS vaccine in Omicron backbone. So even if there is recombination, it would be a defective Omicron virus recombining with a circulating Omicron virus. And no recombinants containing fragments of ancestral virus would be generated. The concept is similar to FluMist, of which polymerase genes are attenuated. In addition, Omicron strain such as BA.5 preferably infects upper respiratory tract, rather than lower respiratory tract in humans. Therefore, the use of Omicron backbone in the second-generation of IBIS will maximise the safety when used in humans. We added a few sentences in the discussion mentioning this idea (Line:427-430).

Comment #8:

- Why does the time between the last vaccination and infection change between 14 and 28 days?

Address #8:

Sorry for the confusion. For all the animal experiments involving viral challenge, the animals were vaccinated with 2 doses of IBIS as follows, 1st dose on day 0, 2nd dose on day 14 and infection on day 28; except for the experiment in Figure 4j in which animals were only vaccinated with 1 dose of IBIS on day 0 and then challenged on day 14. We have updated all the figure legends specifying that infections were performed “14 days post second-dose of vaccination” to avoid confusion.

The reason for using single-dose vaccination in Figure 4j is that we were testing the lower effective dose, in which the subtle difference could possibly be masked by second dose boosting. This is also added in Line:255-257.

Comment #9:

- Would it be possible to get the lung histology scored?

Address #9:

Yes. Histology scoring for the hamster lungs harvested at day 5 post-infection/co-housing has now been included in Supplementary Figure 6e.

Comment #10:

- Please explain what the dark discoloring is in the lungs in Figure S4, IBIS-vac/SARS2.

Address #10:

The dark spots on the H&E-stained slides were clusters of immune cells, resembling the structure of inducible Bronchus-Associated Lymphoid Tissue (iBALT) which have been shown beneficial for fighting acute respiratory viral infections by accelerating immunity to pathogens (reviewed in

Silva-Sanchez and Randall, 2020). Distinct from dispersed perivascular infiltration of lymphocytes, iBALT are lymphoid aggregates situated adjacent to a bronchus, and next to a vein or an artery. Clusters of B cells, T cells, and other supporting cells are present in these structures.

Reference: Silva-Sanchez A, Randall TD. Role of iBALT in Respiratory Immunity. Curr Top Microbiol Immunol. 2020;426:21-43.

We performed an additional immunofluorescence staining on the mouse lung sections, and detected the presence of B220+ cells and CD4+ cells in the cell aggregates we observed. (Supplementary Figure 14). A paragraph in the discussion has also been added to discuss this observation. (Line:393-401)

Minor comments (#11-#31)

Comment #11:

- Please provides stats paragraph in methods. Please note that one cannot use a student's T test in animal studies with smaller numbers. A Mann-Whitney test is more appropriate. I do not think it will change the significance of your studies.

Address #11:

Thanks for the guidance and suggestion. We have now included a stat paragraph in methods; and Mann-Whitney test is now used for the statistical analysis of all animal experiments.

Comment #12:

- Ensure all abbreviations are in the figure legends (e.g. S/N), and all stat abbreviations *, **, ***

Address #12:

Figure legends have been updated accordingly.

Comment #13:

- Ensure spelling is consistent throughout (e.g. delta vs Delta, n=6 vs N=6)

Address #13:

We corrected and updated in the revised manuscript.

Comment #14:

- Please make it clear throughout that IFN β replaced the orf8. this is e.g. not clear in sentence 109 and creates confusion.

Address #14:

We improved the sentence to state that the interferon beta (IFN β) expression cassette was integrated "by replacing orf8 segment in the viral genome". (Line:129-130)

Comment #15

- Please explain why the 90 synonymous changes in the eE transgene in the stable cell line will prevent the recombination (line 138)

Address #15:

Recombination relies on recombination arms that have high homology between 2 sequences. By maximizing the difference between Envelope in the viral genome and the Envelope transgene, the homology becomes minimal. Replication of coronaviruses is compartmentalized, and so the viral genome would unlikely recombine with mRNA in theory. Nevertheless, in the eE transgene the

longest stretch of sequence that is identical to the viral genome is only 5-nucleotide-long (Supplementary Figure 1b), making recombination very unlikely. We added the wordings “to minimize homology” on Line:135.

Comment #16:

- Please specify for all figures and in the text which SARS-CoV-2 variant you use

Address #16:

We have updated the figures and the text specifying the “ancestral”, “Delta” and “Omicron” strain used.

Comment #17:

- Legend Figure 1B; it looks like it can still replicate in cells it infected, just not multiple rounds.

Address #17:

Yes, viral RNA replication could still occur within the individual infected cells because the viral polymerase and the viral components required for RNA replication should be expressed. However, the lack of Envelope protein prevented the assembly of progeny virions. We updated the text and changed “single-round replication” to “single-round infection” to avoid confusion.

Please see Line: 103, 124, 146, 157, 166, 367, and 376.

Comment #18:

- Supplementary Figure S2 is very hard to read.

Address #18:

Sorry for that. We have counted the number of fluorescent foci by ImageJ and added it below each fluorescence image.

Comment #19:

- Is there a way to quantify the fluorescence?

Address #19:

Yes, fluorescent foci were counted by ImageJ and the counts have been added into the figure.

Comment #20:

- Please compare VN titers in Figure 3B to the BNT-vaccinated animals.

Address #20:

We have repeated the FRNT assay together with BNT vaccinated mouse sera and data in Figure 3b have been updated.

Comment #21:

- Line 216: Please explain why IgM plays a particular role in virus neutralization, and do you expect to see this long-term?

Address #21:

Since the titer of IgM was much lower than IgG in the IBIS vaccinated mouse sera, we think it may not play a significant role in virus neutralization. Therefore, we decided to withdraw the IgM data from Figure 3 to avoid misleading/misunderstanding. The text has also been updated (Line: 218). Thanks for the comment again.

Comment #22:

- Experiment Figure 3, please explain why mRNA mice were not challenged also.

Address #22:

We have previous experience on vaccinating mice with the commercial BioNtech BNT162b2 mRNA vaccine. Immunizing mouse with 1 μ g of BNT162b2, which is the same dosage used in current study, was effective in completely protecting the mice. Therefore, the BNT162b2 mRNA-vaccinated group has not been included in the infection experiments.

Comment #23:

- Legend Figure 3A: missing description mRNA mice

Address #23:

Thanks for indicating the mistake. BioNtech BNT162b2 mRNA vaccine was used in this study. The legend of Figure 3 has been updated with the vaccine name, dose, and vaccination route of vaccination.

Comment #24:

- Figure 4A: states day -29 and -15 in figure, but day -28 and -14 in legend. Please clarify.

Address #24:

Sorry for the error. They have been changed to day -28 and day -14 respectively.

Comment #25:

- Line 227: I do not believe you can claim sterilizing immunity; you just have not been able to detect replicating virus. This does not mean that the virus did not replicate somewhere.

Address #25:

Thanks for the comment and agree with that. We have updated the sentence (Line: 37-41, 113-114, and 227-229) and avoid stating “sterilizing immunity”.

Comment #26:

- Please split up Figures 5E and G so they are easier to read.

Address #26:

Thanks for the suggestion. The data points are now separated into lung, nasal turbinate and trachea accordingly. Please see the updated Figure 5e and 5g.

Comment #27:

- The authors do not discuss the pie charts in Figure 7, so I suggest you either remove them or you perform stats on them and discuss them in the text.

Address #27:

As suggested, we have removed the pie charts and categorized the activated CD4⁺ or CD8⁺ T cells into multifunctional, bifunctional and monofunctional at the bottom of the bar charts instead (Lower panels of Figure 7b-7e).

Comment #28:

- Figure 8, specify against what antigen (ancestral?)

Address #28:

It was RBD of the ancestral Spike. Legend of Figure 8 has been updated.

Comment #29:

- Figure 9, please show in log scale, like all others are. Also, the legend states there is a grey dotted line, but I do not see it.

Address #29:

The graph is now plotted in log scale with the grey dotted line (cut-off) shown. Please see the revised Figure 9b. The text was also updated to describe viral titer in log scale (Line: 346-348).

Comment #30:

- I do not think panels S9G and S10G are needed

Address #30:

Thanks for the comment and they are now removed.

Comment #31:

- Please remove panel S11A, since it is also in 7A

Address #31:

Thanks for the comment again and it is now removed accordingly.

Point-to-point address

Reviewer #3:

Comment #1:

1) The authors described IBIS as single-round replicative (line 150, line 160, Fig 1 and S2, etc). Whether IBIS is attenuated because of defect in virus replication or assembly should be carefully verified. In order for immunostaining to work (Fig. 1), the virus has to be replication competent. Otherwise, no viral antigens would be available for staining. If the attenuation is caused by defect in assembly, perhaps electron microscopy or CryoEM can shed light on that. Of note, SARS-CoV that lacks E is attenuated (JVI 2007, p. 1701; JVI 2008, p. 7721).

Address #1:

Thank you for the comment and we have a chance to clarify. It is correct that IBIS did carry out viral RNA replication within the individual infected cells, which is the reason why nucleoprotein could be immuno-stained as pointed out. But it could not complete the replication cycle due to the lack of Envelope protein.

To improve our presentation, we have updated the text and changed all “single-round replication” to “single-round infection” to avoid confusion (Line: 103, 124, 146, 157, 166, 367, and 376). Line 104-105 also states that IBIS “would express IFN β in concert with other viral proteins”.

Comment #2:

2) IBIS appears to be very immunogenic in animal models, which is a bit counterintuitive. If IBIS infection is only single-round, how would vaccination induce measurable amount IFN- β in mouse lungs? The virus would not even be able to arrive at lungs to express IFN- β . Can authors show how responsive IBIS is to IFN β treatment in comparison to wild type virus?

Address #2

Thanks for the comment again. We have updated the text and changed “single-round replication” to “single-round infection”, so as to clarify that IBIS could replicate its RNA genome within the individual infected cells.

Similar to nucleoprotein that was detected by immunostaining (Figure 1b and 1d), single-round infection of IBIS in parental VeroE6 cells could produce a considerable amount of IFN β , which was secreted to the supernatant and became detectable by ELISA (Figure 2c).

In mice, IBIS did not have multiple-round infection and no progeny viruses spread in the lungs after intranasal vaccination. Lung viral titer of replicative SARS-CoV-2 viruses peaked on day 2 post-infection by intranasal inoculation. It reached almost 1×10^7 PFU per gram of lung tissue on our hands (Figure 3g, left column). Yet, none was detected at day 2 after IBIS intranasal inoculation (Figure 3g, right column), suggesting that the IFN β detected in IBIS-vaccinated mouse lungs was not due to multiple-round IBIS infection in the organ.

To rule out the possibility that the IFN β we detected was induced from the host instead of expressed from IBIS, we performed an additional mouse experiment comparing IBIS with its counterpart that does not contain the IFN β cassette (mE) (Supplementary Figure 3). IBIS or mE was intranasally inoculated into mice, and bronchoalveolar lavage (BAL) was collected at 18 hours post-inoculation. In the BAL fluid, IFN β and IP-10 (an IFN-induced chemokine) were only detected in IBIS-vaccinated mice, but not in mE-vaccinated mice. This suggested that IFN β detected in the mouse lungs was produced directly from IBIS.

Description of this result has been added in the text in Line: 197-201.

To summarise,

- 1) Single-round infection of IBIS allowed viral RNA replication and IFN β expression within the individual infected cells.
- 2) IBIS was incapable of multiple-round infection nor spreading in mouse lungs.
- 3) The IFN β detected was expressed directly from IBIS, but not from host response.

Comment #3:

3) Is IFN- β in IBIS an immunogen instead?

Address #3:

We do not consider IFN β in IBIS an immunogen.

IFN β has been widely used clinically in humans for treating various diseases including multiple sclerosis (ref.1) and hepatitis C (ref.2), and showed good safety profile. In the early stage of COVID-19 pandemic, IFN β has also been used in clinical trials to treat patients without adverse effects reported (ref.3). Given that IFN β is only transiently expressed during the single-round IBIS infection, it is unlikely that an autoimmune response against IFN β which is a self-antigen would be generated.

References:

1. M Filipi, S Jack. Interferons in the Treatment of Multiple Sclerosis. *Int J MS Care*. 2020 Jul-Aug;22(4):165-172.
2. D Festi, et al. Safety of interferon β treatment for chronic HCV hepatitis. *World J Gastroenterol*. 2004 Jan 1; 10(1): 12–16.
3. IFN Hung, et al. Triple combination of interferon beta-1b, lopinavir-ritonavir, and ribavirin in the treatment of patients admitted to hospital with COVID-19: an open-label, randomised, phase 2 trial. *Lancet*. 2020 May 30;395(10238):1695-1704.

Comment #4:

4) Similarly, IBIS appeared to offer protection that has not even be achieved in natural infection in that it induces sterile immunity (line 41 and 227). Do the authors know the kinetics of IBIS after vaccination? Is the vaccine virus only detectable in nasal cavity? What is the mechanism for its induction of protection in the lungs if the vaccine virus only infects the nasal cavity?

Address #4:

Thank you for the comment. After careful consideration, we decided to avoid using the term “sterile immunity” in the text. Please see Line: 37-41, 113-114, and 227-229.

To confirm that IBIS does not have multiple-round infection in vivo, we vaccinated K18-hACE2 mice intranasally, and harvested the nasal turbinate and lungs at day 1 and day 2 after vaccination for viral titer quantitation by plaque assay:

No infection virus could be detected in nasal turbinate and lungs at day 1 and day 2 post-vaccination. Since there was no infectious virus on day 2, the time at which infectious SARS-CoV-2 virus multiplies to peak titer, we did not proceed to day 3 and onwards.

Although IBIS vaccine was applied through intranasal route, a strong systemic antibody response has been induced in the animals, indicating a successful immunization. We have monitored the serum antibody level at time-points after vaccination. RBD-specific antibodies could be detected as early as 7 days post-vaccination. The antibody level continued to increase on day 14, during which we boosted the animals with a second dose of IBIS. The antibody titer showed an additional one-log increase at 14 days after boosting, as exemplified in Figure 3c.

This is similar to mRNA vaccination, in which the vaccine given intramuscularly can elicit a strong systemic antibody response and generate protective immunity in mice. IBIS vaccination given intranasally could protect mice from lethal SARS-CoV-2 infection likewise. However, the merit of intranasal route of vaccination is that mucosal immunity is also induced for optimal protection of the respiratory tract, as demonstrated by T cell activation assay (Figure 7). As a result, intranasal IBIS vaccination elicits mucosal and systemic adaptive immunity, protecting the mouse lungs from the intranasally inoculated SARS-CoV-2 virus infection.

Comment #5:

5) In multiple figures, the authors showed viral titers from IBIS-vaccinated animals at levels below detection limit. How was the viral titer determined? By plaque assay or by quantifying the immunostained lung sections?

Address #5:

Regarding the quantitation of viral titer in animal tissues, infectious virus was quantitated by plaque assay and the result was presented as plaque-forming unit per gram (PFU/g) of tissue.

Taking mouse lung as an example, the mouse lung (approximately 0.1g) was harvested and homogenized in 1mL PBS. After centrifugation, the clear supernatant was 10-fold serially diluted for 5 rounds. 100µL of each dilution was then inoculated into each of 6 wells of VeroE6-hTMPRSS2 cells for plaque assay.

Therefore, the detection limit is:

1 PFU/100µL of undiluted lung homogenate, which is equivalent to

10 PFU/mL of undiluted lung homogenate; or

10 PFU/lung which is 0.1g; or

100 PFU/g of lung tissue.

Therefore, a grey dotted line is drawn at 1×10^2 PFU/g to represent the detection limit of viral titer in mouse lungs, as exemplified in Figure 5a-5c.

Thanks for all the comments so that we can further improve our manuscript.

Reviewers' Comments:

Reviewer #1:

Remarks to the Author:

I thank you for carefully reading my comments. I believe they have been sufficiently answered, and I would be happy to support publication of the manuscript. Good luck with your future studies; I look forward to reading about them.

Reviewer #3:

None

Point-by-point response

Reviewer #1:

Comment:

I thank you for carefully reading my comments. I believe they have been sufficiently answered, and I would be happy to support publication of the manuscript. Good luck with your future studies; I look forward to reading about them.

Response:

Thanks for the comments and support.